# TIDES: TRAINING-FREE INSTANCE DETECTION FROM SEMANTICS

## ABSTRACT

Efforts to leverage the coarse semantic understanding of vision-language dual encoder models, such as CLIP, for dense prediction tasks without training have shown promise, particularly in training-free open-vocabulary semantic segmentation (TF-OVSS). However, instance segmentation (TF-OVIS) remains largely unexplored because dual encoder models cannot distinguish individual instances on their own. We systematically evaluate the suitability of promptable segmentation models (PSMs), such as SAM, as sources of accurate instance delineation and present **TIDES** (**T**raining-free **I**nstance **DE**tector from **S**emantics), a pipeline that repurposes any pair of TF-OVSS and PSM for instance segmentation. At its core is our instance-oriented (IO) scoring, which leverages patch-level semantic alignments from TF-OVSS to re-evaluate PSM-generated masks, accurately identifying individual object instances without training, instance-level labels, or external detectors. Extensive evaluation on the MS COCO-based OVIS benchmark across multiple TF-OVSS and PSM combinations demonstrates TIDES' flexibility and effectiveness: it surpasses the previous best TF-OVIS method by 9.2 AP and naive baselines with the original scoring by 2.7 AP.

## 1 INTRODUCTION

Vision-language dual encoder models like CLIP (Radford et al., 2021) have driven major progress in open-world and zero-shot tasks by learning strong semantic representations from coarse supervision (e.g., image–text pairs). Recent studies have shown that this knowledge can also be leveraged for patch-level classification, enabling rapid advances in training-free open-vocabulary semantic segmentation (TF-OVSS) (Li et al., 2025; Wang et al., 2024a; Hajimiri et al., 2025). However, their limited ability to distinguish instances constrains their applicability in many real-world scenarios.

To address this gap, we explore promptable segmentation models (PSMs), such as SAM (Kirillov et al., 2023), as sources of accurate delineation of instances. We derive a novel instance-oriented (IO) scoring method by systematically analyzing the inherent biases of PSMs, and present **TIDES** (**T**raining-free **I**nstance **DE**tector from **S**emantics), a TF-OVIS pipeline that repurposes any pair of PSMs and TF-OVSS models for instance segmentation in a fully training-free manner.

Combining dual encoders with PSMs for training-free instance segmentation remains underexplored, primarily due to two factors: (1) the lack of quantitative analysis of PSMs' limitations in capturing instance-level information, and (2) the common assumption that TF-OVSS methods, which lack instance-level cues, are not directly applicable for instance segmentation.

Studies on (1), the limitations of PSMs, are essential for leveraging their knowledge with minimal side effects. While PSMs often produce visually appealing masks, their highest-confidence predictions frequently focus on distinctive parts rather than complete objects. This reveals a fundamental misalignment with human instance-level perception, as human attention is object-centric rather than driven by isolated visual features (Chen, 2012). In Section 3, we systematically analyze this limitation and show that PSMs are capable of generating masks consistent with human perceptions of object instances, but these correct masks often do not receive the highest confidence scores.

Building on our findings, we derive the IO score, which assigns high values to masks covering individual instances (Section 4). Using patch-level semantic alignment scores from TF-OVSS, each of the PSM's raw mask predictions is first converted into a scale-invariant, semantic-aware

Figure 1: With TIDES, any pair of TF-OVSS and PSM can be repurposed for TF-OVIS. It re-evaluates PSM-generated masks with IO scoring, producing outputs better aligned for instance segmentation.

Figure 2: TF-OVIS qualitative comparison of TIDES and Zip (Shi & Yang, 2024), the previous best.

embedding. These embeddings are then clustered via kernel density estimation to identify correct instance masks and assign appropriate scores. Our qualitative evaluation shows that IO scoring increases the likelihood of selecting the correct instance mask by 22% for SAM, particularly for objects with uniform visual appearance.

Finally, we challenge (2) with our novel TF-OVIS pipeline, TIDES (Section 5). Its key strength is preserving the plug-and-play modularity of its components while being optimized for instance identification. This distinguishes our work from the current state-of-the-art (SOTA), Zip (Shi & Yang, 2024), which is restricted to CNN-based TF-OVSS methods. Our approach also differs from object detector–based OVIS methods, such as GroundedSAM (Ren et al., 2024), which require training with dense annotations, including instance-level masks or bounding boxes paired with text.

Our comprehensive evaluation in Section 6 confirms the impact of IO scoring in TIDES. On the MS COCO-based zero-shot OVIS benchmark (Lin et al., 2014), we observe consistent improvements over naive baselines using the original scoring, with an average gain of 2.7 AP and surpass the previous best, Zip (Shi & Yang, 2024), by 9.2 AP. To guide future work, we conduct additional studies with TIDES (Section 7 and Appendix), quantitatively and qualitatively showing that the technical challenges of TF-OVIS can be decomposed into three sub-problems: achieving fine-grained semantic alignment, identifying in-context objects, and delineating correct instance boundaries.

## 2 RELATED WORKS

### 2.1 TRAINING-FREE OPEN-VOCABULARY SEGMENTATION

**Semantic Segmentation (TF-OVSS).** CLIP-Surgery (Li et al., 2025) is the first work to demonstrate the feasibility of tackling dense prediction tasks using vision-language dual encoders with coarse semantic alignment. It introduces architecture and feature surgery to address misalignment and noisy activations in patch-text correspondence, enabling high-quality Class Activation Maps (CAMs) extraction from CLIP without fine-tuning or additional supervision.

Building on this foundation, various approaches have been proposed to tackle semantic segmentation in a training-free setting (TF-OVSS). Methods like SCLIP (Wang et al., 2024a), ClearCLIP (Lan et al., 2024a), NACLIP (Hajimiri et al., 2025), and CLIPtrace (Zhu et al., 2024) enhance patch-text correspondence by modifying self-attention mechanisms: replacing query-key cross-correlation with query-query, key-key, or value-value self-correlations. Additional denoising strategies include ClearCLIP removing residual connections and feed-forward layers (Lan et al., 2024a), NACLIP

applying local Gaussian filtering to smooth activations (Hajimiri et al., 2025), ResCLIP performing cross-correlation-based alignment adjustments with iterative refinement (Yang et al., 2024), CLIPtrace clustering similar patch-text correspondences (Zhu et al., 2024), and SC-CLIP (Bai et al., 2024) and CLIPSeg (Liu et al., 2025) using intermediate features to locate and correct noisy activations.

Another line of work focuses on improving semantic alignment using additional pretrained models: vision foundation models (ProxyCLIP (Lan et al., 2024b), Trident (Shi et al., 2024), CorrCLIP (Zhang et al., 2024a), DBA-CLIP (Yang & Gong, 2024), FreeDA (Barsellotti et al., 2024), kNN-CLIP (Gui et al., 2024)), diffusion models (FreeDA (Barsellotti et al., 2024), FreeSeg-Diff (Corradini et al., 2024)), and image captioning models (FreeSeg-Diff (Corradini et al., 2024)).

**Instance Segmentation (TF-OVIS).**  While denoising methods have shown notable improvements in patch-level semantic alignment, few have explored training-free strategies for identifying patches associated with distinct instances. Zip (Shi & Yang, 2024) pioneers this direction by framing TF-OVIS as annotation-free instance segmentation, clustering ResNet-based CLIP features to localize object boundaries. However, its assumption that patches representing object boundaries are spatially scattered with hollow interiors does not generalize reliably, as shown in our studies in Section 6.

## 2.2 PSM: PROMPTABLE SEGMENTATION MODELS

PSMs refer to models that generate masks for regions indicated by spatial prompts such as points, boxes, or masks, with SAM (Kirillov et al., 2023) being the most successful example.

Targeting common object segmentation, subsequent work has advanced in two main directions:

**Efficiency-oriented.** SAM's high memory and computational cost has introduced many difficulties in production, leading to various efforts to reduce its overhead (Sun et al., 2024). Notable strategies include knowledge distillation (EdgeSAM (Zhou et al., 2023), MobileSAM (Zhang & Jiao, 2023), EfficientSAM (Xiong et al., 2024)), alternative network architectures (FastSAM (Zhao et al., 2023), EfficientViTSAM (Sun et al., 2024)), and model compression techniques like quantization and pruning (TinySAM (Shu et al., 2025), SlimSAM (Chen et al., 2024b)).

**Quality-oriented.** Meanwhile, a group of innovations focuses on improving segmentation quality, especially around object boundaries and fine details. SAM2 (Ravi et al., 2024) enhances the decoder with dual-path attention for sharper masks. UnSAM (Wang et al., 2024c) uses self-training to emphasize perceptual coherence by modeling local feature similarity. SegNext (Liu et al., 2024a) proposes dense representations of diverse visual prompts to better interpret input prompts. HQ-SAM (Ke et al., 2023) explores plug-and-play enhancements to SAM variants by explicitly training a learnable token to generate high-quality masks.

## 2.3 COMBINING DUAL ENCODERS WITH PSMs FOR OVIS

Box prompt support in PSMs enables seamless conversion of open-vocabulary object detectors into instance segmentation solutions (Zhu & Chen, 2024; Feng et al., 2025). A prominent example is GroundedSAM (Ren et al., 2024), which combines GroundingDINO (Liu et al., 2024b) with SAM.

Recent approaches employing vision-language dual encoders for instance localization rely on additional training to remap extracted features into a new space (SAM-CLIP (Wang et al., 2024b) and FrozenSeg (Chen et al., 2024a)). The only training-free method is Zip (Shi & Yang, 2024), yet it shows limited usability because it is tied to CNN-based TF-OVSS.

## 3 UNDERSTANDING PSM'S BIAS TOWARD LOCAL VISUAL FEATURES

Human visual attention is directed toward perceptual objects rather than disconnected regions (Chen, 2012). Replicating such human perceptual capabilities in computational models requires careful consideration of these object-centric biases (Tai et al., 2025; Roh et al., 2025).

When asked about the gap between PSMs' perception and human perception, one might acknowledge its existence, often citing PSMs' tendency toward oversegmentation. However, the magnitude of this gap remains unclear due to the lack of systematic analysis. In this section, we present the first

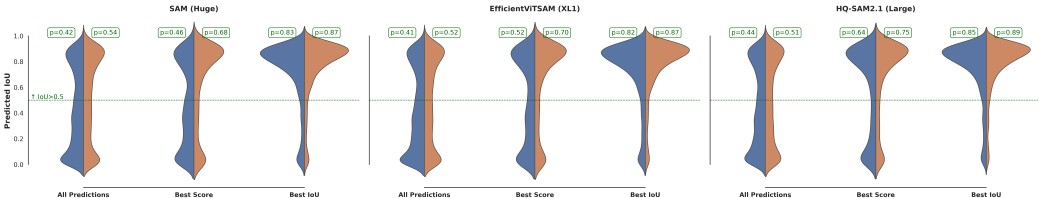

Figure 3: Distributions of raw predictions from SAM (Kirillov et al., 2023), EfficientViTSAM (Zhang et al., 2024b), and HQ-SAM2.1 (Ke et al., 2023; Ravi et al., 2024) across IoU values for *uniform* and *non-uniform* objects. $p$ indicates percentage of predictions with IoU > 0.5.

rigorous and quantitative study of PSMs' misalignment with human perception. This analysis not only provides a theoretical foundation for our IO scoring method but also offers actionable insights for adapting PSMs to other instance-level tasks.

### 3.1 ANALYSIS SETUP

PSMs consist of an image encoder, a prompt encoder, and a mask decoder. Given an input image and a prompt referring to a target region in the image, the encoders first generate image and prompt embeddings. The mask decoder then processes these to produce $m$ candidate masks (typically $m = 3$), each with a confidence score indicating alignment with the prompt. A fixed-value threshold and non-maximum suppression (NMS) can then be leveraged to eliminate redundant predictions.

To quantify the perception gap between PSMs and humans, we use an instance segmentation dataset, specifically the MS COCO 2017 validation set (Lin et al., 2014). For each labelled instance, we randomly select $n = 5$ points per instance and collect pre-filtering (raw) outputs of a PSM for each point. We then analyze the distribution of Intersection over Union (IoU) between the ground truth (GT) masks and the following groups: (1) *All Predictions* ($m \times n$ per instance), (2) *Best Score*: the highest-confidence prediction per prompt ($n$ per instance), and (3) *Best IoU*: the highest-IoU prediction per prompt ($n$ per instance).

To better characterize the bias, we divide the 80 categories into two groups: *uniform* and *non-uniform*. The *uniform* group includes 62 categories with relatively homogeneous appearance, consistent texture, and simple shapes, where the entire object is often perceived as a single, cohesive entity (e.g., *apple*, *dog*, *donut*), while the *non-uniform* group includes 18 categories (by contrast) composed of multiple, visually distinct parts or texture (e.g., *car*: wheels and body, *person*: head, clothing, and limbs). The complete list for each group can be found in the Appendix (Section A.3).

In this section, we primarily focus on the largest architectures of three representative models in the field of PSM: SAM (Kirillov et al., 2023), EfficientViTSAM (Zhang et al., 2024b) (the most efficient variant), and HQ-SAM (Ke et al., 2023) applied on top of the latest SAM variant (SAM 2.1 (Ravi et al., 2024)), which we refer to as HQ-SAM2.1 (the highest quality segmentation). Studies with other PSMs are also available in the Appendix (Section A.3).

### 3.2 FINDINGS

As shown in the *All Predictions* graphs for each model (Figure 3), the three models exhibit similar IoU distributions, characterized by a bimodal pattern with peaks in the high-IoU region ($0.8 \sim 1.0$) and the low-IoU region ($0.0 \sim 0.2$). However, object appearance does have an effect: *uniform* objects are generally easier to segment, as their IoU distribution in the high-IoU region (IoU > 0.5) is slightly higher than that of *non-uniform* objects.

The proportion of well-aligned masks increases slightly when restricted to the highest-confidence predictions (*Best Score*), indicating that PSMs capture some understanding of individual instances. However, comparing this distribution to that of the best IoU (*Best IoU*) reveals that the highest-confidence scores are not fully aligned with the underlying instances.

Overall, our findings can be summarized as follows: (1) PSMs often generate masks aligned with human perceptions of object instances—not just visual boundaries—but (2) do not necessarily assign the highest scores to these masks.

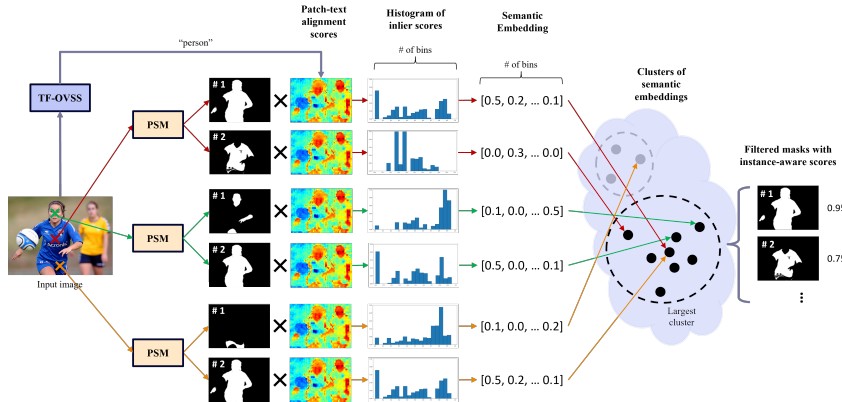

Figure 4: Overview of IO scoring: Raw PSM predictions from multiple prompts are aggregated based on their semantic alignment with the underlying instance, and cluster statistics are subsequently used to quantify instance alignment.

## 4 DESIGNING SCORES THAT REFLECT INSTANCE UNDERSTANDING

Given the high proportion of masks with IoU $> 0.5$ in the *Best IoU* distribution, the likelihood that at least one of the $m$ raw masks aligns well with the underlying instance is also high. This finding not only confirms that PSMs can be sources of accurate instance delineation but also motivates the need for a new score that better reflects a mask's alignment with the underlying instance. Ideally, the new score should close the gap between the *Best Score* and *Best IoU* distributions.

At a high level, we run a PSM with multiple prompts on an object, aggregate all raw predictions, analyze their similarities using patch-text alignment scores computed with TF-OVSS, and derive a score representing how well each mask covers the underlying instance (Figure 4). To guide the reader through the evolution of the idea, we first assume access to a set of points belonging to the same instance (single-instance case; Section 4.1) and then gradually relax this assumption to show that the approach extends to multi-instance cases (Section 4.2). Finally, we quantitatively verify the soundness of our approach in Section 4.3.

### 4.1 IDENTIFYING THE CORRECT MASK FOR A SINGLE INSTANCE

By accumulating all point prompts referring to the same instance into a batch, we can run a PSM and collect all the raw masks it generates. We then construct a semantic embedding for each mask using the following process: (1) compute patch–text alignment scores using a TF-OVSS method; (2) select foreground patches by thresholding per-patch min–max normalized scores at 0.1; (3) extract their corresponding scores and construct a histogram over them; and (4) normalize the histogram by the number of patches to make it scale-invariant. Since semantically identical regions yield similar alignment scores, masks covering the same instance should produce similar distribution. In other words, concatenating the normalized bin values yields a fixed-length, scale-invariant semantic embedding for each mask.

Such a conversion of masks into embeddings allows masks covering semantically identical regions to be closely located in the embedding space. Furthermore, given the high likelihood of observing the correct mask among the raw predictions (Section 3.2), raw masks for the same instance form dense clusters, with the densest regions corresponding to the masks that are most aligned with the underlying instance. In this context, the local density—measuring how many other points lie nearby—serves as a natural proxy for alignment with human perception of the instance.

Specifically, let $\{\mathbf{x}_i\}_{i=1}^N \subset \mathbb{R}^B$ be the generated embeddings, where $N$ is the number of masks/embeddings and $B = 40$ is the number of bins. The IO scoring function, $f_{\text{IO}}$, for a mask $\mathbf{x}_i$ is defined identically to the local density calculation:

$$f_{\text{IO}}(\mathbf{x}_i) = \frac{1}{N} \sum_{j=1}^N \exp\left(-\frac{\|\mathbf{x}_i - \mathbf{x}_j\|^2}{2h^2}\right),$$

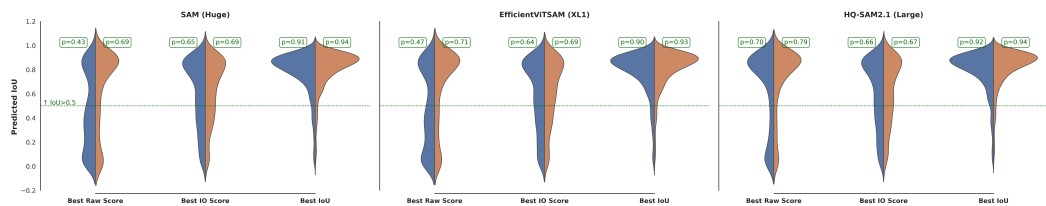

Figure 5: Distributions of highest original scores (*Best Raw Score*), highest IO scores (*Best IO Score*), and highest IoU predictions (*Best IoU*). The two graphs in each figure correspond to *uniform* and *non-uniform* objects. $p$ indicates percentage of predictions with IoU > 0.5.

where $h = 0.5$ is the bandwidth hyperparameter and the resulting values lie in the range $\left(\frac{1}{N}, 1\right]$.

KDE uses this local density non-parametrically to identify cluster centers, making it particularly well-suited for our use case. Our implementation uses a density threshold of $0.90$.

## 4.2 IDENTIFYING CORRECT MASKS FOR MULTIPLE INSTANCES

The target task, instance segmentation, requires identifying individual objects. Therefore, assuming access to point prompts from the same instance is unrealistic. However, using semantic segmentation models, points referring to the same class (though not necessarily the individual instance) can be derived. Given that our semantic embeddings are scale-invariant (Section 4.1), embeddings from different instances still form coherent clusters, provided they correspond to semantically identical objects. In theory, our IO scoring remains valid for multiple-instance cases, provided that TF-OVSS captures the underlying semantics correctly.

## 4.3 IMPACT OF THE INSTANCE-ORIENTED SCORE

Figure 5 illustrates the evolution of the *Best Score* distribution under the refined settings across the three models: SAM, EfficientViT-SAM, and HQ-SAM2.1. First, *Best Raw Score* refers to the highest original score predictions from $n$ random point prompts within GT masks. Next, *Best New Score* corresponds to the highest IO-score predictions from $n$ point prompts selected from foreground regions identified with a TF-OVSS method, CLIP-Surgery (Li et al., 2025). Specifically, per-class min–max normalized patch–text scores are softmaxed against alignment scores with `background` text, and those greater than 0.1 are considered foreground. Lastly, *Best IoU* represents the highest-IoU predictions from $n$ random point prompts within GT masks, included as a reference to indicate the upper bound of improvement.

It is worth noting that the curves for *Best Score* and *Best IoU* may differ from those in Figure 3, since Figure 5 summarizes per-instance distributions (best out of $n \times m$), rather than per-prompt distributions (best out of $m$ for each of the $n$ point prompts independently).

As summarized in Section 3.2, raw scores for *non-uniform* instances are suboptimal for instance segmentation in both SAM and EfficientViTSAM, with only 30% and 42% of predictions achieving IoU > 0.5, respectively. Our IO score addresses this, increasing these percentages to 59% and 64% (improvements of 29% and 17%). For instances where original scores already show strong instance awareness, IO scoring trades a slight reduction in high-IoU predictions for a substantial decrease in low-IoU ones, transforming the typically bimodal distribution into a unimodal one concentrated in the high-IoU region. By producing distributions skewed toward high-IoU regions for both *uniform* and *non-uniform* objects without additional training, IO scoring mitigates PSMs' bias toward local visual features, making them suitable for instance segmentation.

## 5 TIDES: TRAINING-FREE INSTANCE DETECTOR FROM SEMANTICS

Here, we introduce TIDES, a novel and flexible pipeline that combines TF-OVSS and PSM using IO scoring to achieve instance segmentation without instance-level labels or additional training. Its novelty lies in its generality, being applicable to any semantic segmentation module, including those aligning modalities beyond text with 2D images. As illustrated in Figure 6, the pipeline proceeds

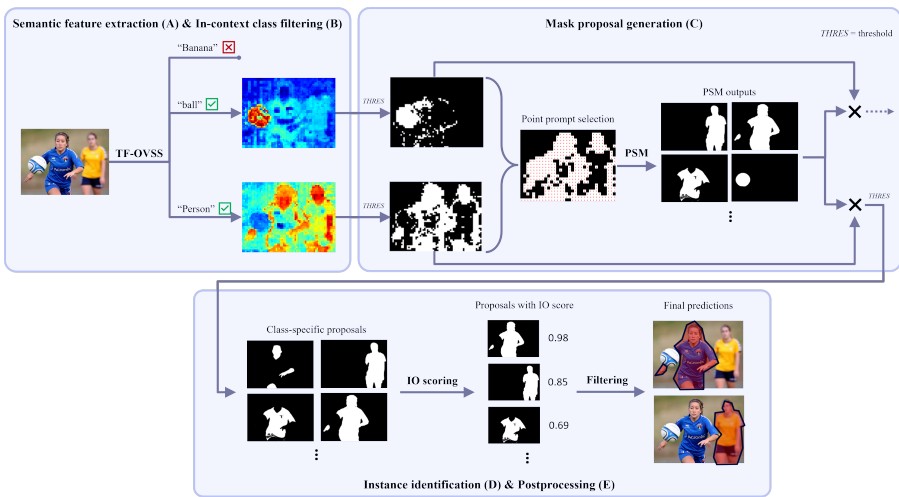

Figure 6: The complete TIDES pipeline from semantic feature extraction to postprocessing.

in five steps: (A) Semantic feature extraction: compute image-text and patch-text alignment scores, (B) In-context class filtering: identify relevant classes among those queried, (C) Mask proposal generation: generate mask proposals for each in-context class, (D) Instance identification: select correct instance masks using IO scores, and (E) Postprocessing: remove remaining invalid masks. Additional diagrams and descriptions are provided in Appendix A.1.

**Semantic feature extraction (A) & In-context class filtering (B).** In a standard dual-encoder setting, the cosine similarity between the [CLS] token and a text embedding represents global (image–text) alignment, whereas the similarity between patch embeddings and a text embedding captures local (patch–text) alignment. TF-OVSS methods focus on amplifying patch–text alignment without additional training to derive precise semantic segmentation, while keeping the global alignment unchanged. To capture fine-grained details, we resize the input image to have a longer side of $512$, then process it at three scales using a sliding-window approach with window sizes $112$, $168$, and $224$, and a stride of half the window size. Patch–text scores from all scales are averaged per pixel and converted back to patch-level using the greatest common divisor of the three patch sizes.

In open-vocabulary settings, identifying which classes are present is crucial. While few-shot detection benchmarks (Nguyen & Todorovic, 2019; Zhang et al., 2023; Everingham et al., 2010) assume prompts always refer to valid objects, this assumption does not hold in text-based zero-shot settings. To filter out irrelevant classes, we introduce a background class to model the absence of targets. A class is retained if its image–text score from any window exceeds that of background.

**Mask Proposal Generation (C).** We first identify foreground patches as described in Section 4.3: normalized patch–text scores are thresholded at $0.1$ for each in-context class. Foreground patches are then aggregated across classes, and any patch marked as foreground for at least one class is used to define a point prompt at its center. These points are passed to PSM as a batch to generate a set of candidate masks. Each generated mask is then compared with the class-specific foreground patches, and if its IoU exceeds $0.2$, it is assigned to that class as a class-specific mask proposal.

**Instance identification (D) & Postprocessing (E).** Subsequent steps operate per class. From the set of proposals, we retain masks from the largest cluster following the process in Section 4. Non-overlapping masks are selected by applying NMS on the IO score with an IoU threshold of $0.7$. Finally, we compute a semantic score for each mask by averaging foreground patch–text scores (min–max normalized), discarding values below $0.1$ to suppress noise.

## 6 EXPERIMENTS

In this section, we demonstrate the flexibility of TIDES by converting various combinations of TF-OVSS and PSM into a TF-OVIS pipeline and evaluating on zero-shot scenarios.

Table 1: Zero-shot TF-OVIS performance of TIDES with different TF-OVSS and PSM combinations on MS COCO 2017. Each reports performance with the original PSM score (Raw) and performance with IO scores. "SS perf." denotes the average TF-OVSS performance on standard benchmarks.

| TF-OVSS (SS Perf.) | PSM | Score | $AR_{10}$ | $AR_{100}$ | $AP_s$ | $AP_m$ | $AP_l$ | $AP_{75}$ | $AP_{50}$ | AP ↑ | vs. Raw |
|---|---|---|---|---|---|---|---|---|---|---|---|
| CS (37.2) | SAM (Huge) | Raw | 40.0 | 41.7 | 5.7 | 25.9 | 37.4 | 14.1 | 22.9 | 14.6 | |
| | | IO | 39.8 | 40.9 | 6.7 | 27.6 | 39.2 | 18.7 | 29.4 | 19.2 | + 4.6 |
| | E-SAM (XL1) | Raw | 39.4 | 40.3 | 6.4 | 27.9 | 40.3 | 17.8 | 27.3 | 18.2 | |
| | | IO | 38.5 | 39.3 | 6.9 | 27.6 | 40.8 | 20.5 | 31.2 | 21.0 | + 2.8 |
| | HQ-SAM2.1 (Large) | Raw | 25.6 | 25.7 | 3.4 | 15.4 | 31.8 | 12.3 | 25.4 | 14.6 | |
| | | IO | 25.9 | 25.9 | 3.5 | 15.8 | 31.8 | 12.5 | 26.3 | 15.0 | + 0.4 |
| SCLIP (39.1) | SAM (Huge) | Raw | 36.4 | 39.2 | 2.6 | 20.9 | 40.8 | 13.2 | 20.9 | 13.7 | |
| | | IO | 35.8 | 37.9 | 3.0 | 21.6 | 41.5 | 17.6 | 26.7 | 18.2 | + 4.5 |
| | E-SAM (XL1) | Raw | 35.2 | 36.8 | 3.1 | 22.7 | 40.9 | 15.2 | 23.4 | 15.7 | |
| | | IO | 34.3 | 35.5 | 3.4 | 23.2 | 41.8 | 18.9 | 28.1 | 19.4 | + 3.7 |
| | HQ-SAM2.1 (Large) | Raw | 22.2 | 22.2 | 1.6 | 12.1 | 30.4 | 11.8 | 22.4 | 13.3 | |
| | | IO | 21.8 | 21.8 | 1.7 | 12.2 | 30.5 | 12.4 | 23.1 | 13.9 | + 0.6 |
| SC-CLIP (43.9) | SAM (Huge) | Raw | 40.0 | 42.5 | 4.3 | 25.1 | 40.4 | 14.6 | 23.7 | 15.3 | |
| | | IO | 38.2 | 40.0 | 4.7 | 25.1 | 39.6 | 18.5 | 28.8 | 19.1 | + 3.8 |
| | E-SAM (XL1) | Raw | 38.6 | 40.1 | 5.1 | 26.8 | 40.6 | 16.8 | 26.0 | 17.4 | |
| | | IO | 37.1 | 38.2 | 5.5 | 26.5 | 40.0 | 20.1 | 30.2 | 20.6 | + 3.2 |
| | HQ-SAM2.1 (Large) | Raw | 23.3 | 23.3 | 2.1 | 14.4 | 30.8 | 12.2 | 23.5 | 14.0 | |
| | | IO | 23.0 | 23.0 | 2.2 | 14.9 | 31.5 | 13.2 | 24.5 | 14.8 | + 0.8 |

## 6.1 EXPERIMENT SETUP

We combine three different TF-OVSS methods—CLIP-Surgery (Li et al., 2025), referred to as CS (on which the previous best TF-OVIS method, Zip (Shi & Yang, 2024), is based), SCLIP (Wang et al., 2024a) (noted for its simple yet effective self-correlation modifications), and SC-CLIP (Bai et al., 2024) (the current state-of-the-art TF-OVSS method)—with three PSMs trained with different objectives: SAM (Kirillov et al., 2023), EfficientViT-SAM (Zhang et al., 2024b), referred to as E-SAM, and HQ-SAM2.1 (Ke et al., 2023; Ravi et al., 2024). This results in nine different combinations.

We follow the standard zero-shot OVIS setup, the MS COCO 2017 validation set (Lin et al., 2014) (5K images, 80 categories), and report mask Average Precision (AP) and Average Recall (AR).

**Baselines.** To evaluate the effect of IO scoring, we first compare TIDES with a version using the original PSM scores, where the instance identification step (D) is skipped (Raw). We also provide a comparison with the recent zero-shot TF-OVIS method, Zip (Shi & Yang, 2024), which consists of three stages: semantic heatmap generation using TF-OVSS (CS), object boundary detection via clustering ResNet-based CLIP local features, and SAM-based mask refinement. Specifically, Zip feeds the original $512 \times 512$ image to the CS module and an upscaled $2048 \times 2048$ image to CLIP RN50x64 to extract local features. The authors further compare Zip with a simpler baseline that runs SAM on all points and uses CLIP to filter relevant masks; we refer to this baseline as SAM+CLIP and include it in our study. All TF-OVSS methods in these experiments use CLIP with ViT-B backbones.

## 6.2 EXPERIMENT RESULTS

Table 1 summarizes how our IO scoring improves TF-OVIS performance over the original scores in the zero-shot setting. On average, TIDES results in a 2.7 AP improvement across all combinations. In general, we observe that the PSM plays a larger role than the TF-OVSS method in achieving high TF-OVIS performance: TIDES with E-SAM consistently achieve the highest scores. The performance gain introduced by the new scoring also aligns with our analysis in Section 4.3: SAM and E-SAM show average increases of 4.3 AP and 3.2 AP, respectively, while HQ-SAM2.1 shows only a minimal increase of 0.6 AP. Our evaluation—including MobileSAM (Zhang & Jiao, 2023), SAM2 (Ravi et al., 2024), and different architectures of the models—confirms our new scoring consistently improves performance over original scoring across PSMs (Section A.6 of Appendix).

As summarized in Table 2, prior efforts to decompose the coarse semantic information of dual encoders for instance segmentation in a training-free manner have shown modest performance; the previous best model, Zip, which reports an AP of 11.8, underscores the difficulty of the task. Nevertheless, TIDES, based on the same PSM (SAM) and a single TF-OVSS run (only with ViT-B variant of CS), outperforms Zip by 3.4 points. Our best combination, CS with E-SAM, sets a new state of the art with an AP of 21.0, surpassing Zip by 9.2 points.

Table 2: Comparison of TIDES with other zero-shot OVIS methods. Performances of SAM-CLIP and GroundedSAM are also included to illustrate the benefit of training and using an explicit detector.

| Method | TF-OVSS & PSM | Training free | $AR_{10}$ | $AR_{100}$ | $AP_s$ | $AP_m$ | $AP_l$ | $AP_{75}$ | $AP_{50}$ | AP ↑ | vs. SOTA |
|---|---|---|---|---|---|---|---|---|---|---|---|
| SAM+CLIP (Shi & Yang, 2024) | CLIP & SAM (B) | ✓ | 10.1 | 36.9 | 1.4 | 3.3 | 5.9 | 2.7 | 4.6 | 2.6 | |
| Zip (Shi & Yang, 2024) | CS & SAM (B & RN) | ✓ | 21.9 | 31.2 | 4.8 | 13.1 | 20.1 | 12.0 | 20.0 | 11.8 | |
| Naive | CS & SAM (B) | ✓ | 34.9 | 36.1 | 4.6 | 22.4 | 34.4 | 12.3 | 21.7 | 13.0 | + 1.2 |
| TIDES | CS & SAM (B) | ✓ | 34.1 | 35.0 | 4.6 | 21.5 | 34.1 | 14.2 | 24.8 | 15.2 | + 3.4 |
| TIDES | CS & E-SAM (XL1) | ✓ | 38.5 | 39.3 | 6.9 | 27.6 | 40.8 | 20.5 | 31.2 | **21.0** | **+ 9.2** |
| SAM-CLIP (Wang et al., 2024b) | CLIP & SAM (B) | ✗ | - | - | - | - | - | - | - | 40.9 | |
| GroundedSAM (Ren et al., 2024) | GroundingDINO (B) & SAM (B) | ✗ | - | - | 33.2 | 57.1 | 68.4 | 53.1 | 70.1 | 52.0 | |

Table 3: Effect of replacing TIDES components with GT counterparts. Class IoU denotes the IoU between predicted and GT classes.

| GT semantic alignments (A) | GT vocabulary (B) | GT instance mask (C/D) | class IoU ↑ | | AP ↑ | |
|---|---|---|---|---|---|---|
| ✗ | ✗ | ✗ | 19.0 | | 20.5 | |
| ✓ | ✗ | ✗ | 85.3 | + 66.3 | 57.4 | + 36.9 |
| ✗ | ✓ | ✗ | 93.0 | + 74.0 | 52.9 | + 32.4 |
| ✗ | ✗ | ✓ | 59.1 | + 40.1 | 68.4 | + 47.9 |
| ✓ | ✓ | ✓ | 98.4 | + 79.4 | 98.4 | + 77.9 |

Figure 2 illustrates the differences between TIDES and Zip on both single- and multi-class images, including cases where the target class is absent. As shown in the first two columns, TIDES performs comparably to Zip in single-class scenarios. In contrast, on multi-class images (last three columns), Zip produces erroneous outputs, whereas TIDES accurately segments the target class and suppresses output when the class is absent. Our in-depth quantitative and qualitative analysis of TIDES and Zip in Section A.7 of the Appendix confirms that TIDES' main advantage lies in its ability to identify in-context classes and distinguish correct instances from background via its IO scoring.

# 7 EXPLORING THE KEY CHALLENGES IN TF-OVIS WITH TIDES

While TIDES achieves notable improvements on TF-OVIS tasks, its 21.0 AP highlights remaining challenges. To better understand the remaining challenges, we analyze TIDES's performance when its subcomponents are replaced with optimal alternatives: (A) semantic feature extraction replaced by a binary semantic mask formed from GT semantic masks, (B) in-context class filtering replaced with GT classes, and (C/D) mask proposal generation and instance identification replaced by GT instance masks. Evaluation is performed on the best TIDES pipeline (CS & E-SAM) using a 500-sample large-object subset of COCO.

Table 3 shows the changes in class IoU (between predictions and GT) and AP after replacing components with GT. As expected, using GT classes instead of in-context class filtering (B) yieds the largest improvement in class IoU (+74.0) while using GT instance masks instead of PSM (C/D) yield tje largest AP gain (+47.9). Deriving semantic alignments using GT masks (A) significantly improves both metrics: IoU (+66.3) and AP (+36.9), emphasizing the importance of accurate semantic localization. When replacing all three components with optimal counterparts, both class IoU and AP approach 100 confirming them as primary bottlenecks.

We highlight the significant performance improvement with optimal semantic alignments. Since TIDES converts any semantic segmentation module (trained or training-free and open or closed-vocabulary) into an instance segmentation pipeline, it will benefit from future advancements in semantic segmentation to enhance instance-level performance.

# 8 CONCLUSION

This work demonstrates that while PSMs often produce instance-aligned masks, they fail to rank them highest. By leveraging the semantic understanding of TF-OVSS, we rescore PSM-generated masks to prioritize those with strong instance correspondence. Building on this, we present TIDES, a flexible pipeline that repurposes a TF-OVSS and PSM pair for TF-OVIS. Our comprehensive analysis confirms the superior performance and flexibility of TIDES in deriving instance segmentation from models trained solely with coarse labels.

REPRODUCIBILITY STATEMENT

To enable readers to reproduce our pipeline, we provide a detailed description of the theoretical and mathematical derivation of our novel IO score (Sections 3 and 5), along with complete implementation details (Sections 5 and A.1). We also analyze the effect of hyperparameter changes (Section A.2) to help readers assess any differences between their implementation and ours.

For reproducing experimental results, we provide the full experimental setup and results (Section 6), along with additional qualitative and quantitative evaluations (Sections A.7 and A.6) to allow verification of reproduced results.

Furthermore, we plan to release the full source code upon acceptance. We believe that the comprehensive descriptions and analyses here provide sufficient guidance for reproducing our study, with any remaining gaps fully addressed by the released code.

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
