# A   APPENDIX

## A.1   IMPLEMENTATION DETAILS

This section supplements the pipeline description in Section 5 with additional implementation details that could not be included earlier due to space constraints. *Italicized* text denotes tunable hyperparameters, with sensitivity analyses provided in Sections A.2 and A.4.

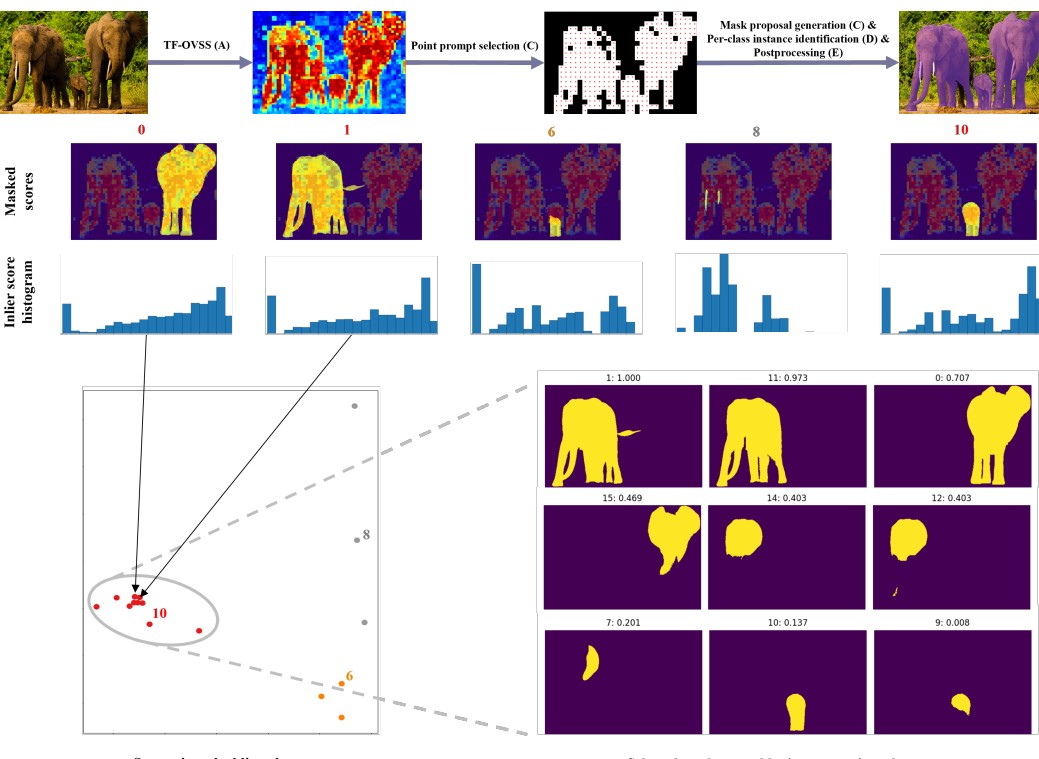

Figure 7: Sample outputs from each step of TIDES. **Top:** Overall pipeline — input image, patch-text semantic alignment scores, foreground patches & selected point prompts for mask generation, and final output. **Middle:** Sample predictions after PSM postprocessing with relaxed criteria and the corresponding histograms. **Bottom:** Semantic embedding clusters (colors denote different clusters). The masks on the right are from the largest cluster, sorted by instance-oriented score. Final postprocessing filters these masks, yielding the outputs shown in the top row.

**Semantic feature extraction (A).**   Following standard dual encoder usage, alignment scores between text and image embeddings are computed via cosine similarity, yielding values in $[-1, 1]$. The input image is first resized to have a longer side of $512$, then we adopt a sliding-window approach with multiple *window scales* and *sizes*, using a stride equal to half the *window size*. Patch–text scores from all windows are upsampled to the original resolution, averaged across scales to form pixel–text scores, and converted back to patch–text scores using the greatest common divisor of patch sizes. This allows TIDES to use a patch size different from that of the dual encoder and improves its ability to capture fine-grained detail.

**In-context class filtering (B).**   Sliding-window inference yields multiple `[CLS]` tokens per image. For each window, we compute the similarity between the `[CLS]` token and target class tokens, softmaxed against the `background` token. A class is marked present if any window produces a score greater than that of `background`.

**Point prompt selection and mask proposal generation (C).**   Foreground patches are identified by taking the patch-wise product of per-class min–max normalized scores and softmax-normalized scores

against background, followed by thresholding at the *foreground region threshold*. All foreground patches across classes are then collected as a batch and passed to the PSM with a single point prompt at the center of each patch. Using every raw PSM prediction significantly increases computation without improving results, so we relax PSM's postprocessing by lowering the *stability score threshold* and raising the *box NMS threshold* to retain diverse masks at lower latency (The exact trade-offs are discussed in Section A.4). Among the generated masks, those whose IoU with the corresponding foreground region falls below the *foreground-mask IoU threshold* are discarded.

**Per-class instance identification (D).** For each mask, patch–text alignment scores from its foreground patches are collected, binned into a histogram (*bin size*), and normalized by count to produce fixed-length semantic embeddings. KDE-based clustering is then applied using *bandwidth* and *density threshold* as hyperparameters. Only masks from the largest cluster are kept, filtering out partial or incorrect predictions. The local density of each selected mask is used as its IO score.

**Postprocessing (E).** Overlapping masks are removed by applying NMS with the IO score and an IoU threshold (*NMS threshold*). Each mask is then assigned a semantic score by averaging its min–max normalized patch–text alignment scores, and those below the *semantic score filtering threshold* are discarded to suppress noise.

## A.2 HYPERPARAMETER SENSITIVITY ANALYSIS

Table 4 reports the range of values explored for each hyperparameter along with the corresponding sensitivity of the pipeline, summarized using mean, median, minimum, and maximum AP. The **bolded** value indicates the setting that achieved the highest performance for each hyperparameter.

The search and analysis are conducted using a pipeline built with CS (ViT-B/16) (Li et al., 2025) and EfficientViTSAM (XL1) (Sun et al., 2024) on a subset of the MS COCO 2017 validation set (Lin et al., 2014), selected to include at least three instances of each of the 80 categories. Empirically, the best-performing configurations on this subset are reasonably consistent with those observed on the full validation set of 5,000 samples.

Table 4: Impact of hyperparameter choices on performance. `[a:b:c]` denotes a range from `a` to `c` (inclusive) with step size `b`. The **bolded** choices indicates the one that led to the best performance.

| Hyperparameter | Stage | Choices (**selected**) | AP | | | |
| --- | --- | --- | --- | --- | --- | --- |
| | | | Minimum | Median | Maximum | Mean |
| *window scales* | A | 1, 2, **3** | 14.8 | 25.9 | 30.2 | 25.3 |
| *window sizes* | A | 56, **112, 168, 224**, 336, 448 | | | | |
| *PSM box IOU threshold* | C | Section A.4 (**0.80**) | 26.6 | 28.2 | 30.1 | 28.0 |
| *PSM stability score threshold* | C | Section A.4 (**0.90**) | | | | |
| *foreground region threshold* | C | [0.10:0.10:0.90] (**0.10**) | 22.8 | 30.2 | 30.4 | 29.3 |
| *foreground-mask IoU threshold* | C | [0.10:0.10:0.90] (**0.20**) | 25.3 | 29.4 | 30.7 | 29.0 |
| *histogram bin size* | D | **0.025**, 0.050, 0.100 | 10.7 | 21.9 | 31.3 | 21.6 |
| *clustering bandwidth* | D | [0.50:0.20:1.50] (**0.50**) | 10.6 | 22.1 | 30.3 | 21.9 |
| *clustering density threshold* | D | 0.75, 0.80, 0.85, **0.90** | 11.6 | 22.4 | 31.6 | 22.1 |
| *NMS threshold* | E | [0.10:0.10:0.90] (**0.70**) | 27.5 | 29.6 | 30.4 | 29.5 |
| *semantic score filtering threshold* | E | [0.10:0.10:0.90] (**0.10**) | 1.6 | 29.1 | 30.3 | 23.3 |

## A.3 ADDITIONAL STUDIES ON PSM BEHAVIOR AND THE IMPACT OF IO SCORING

**Per-Prompt Prediction Distributions of Different PSMs.** Figure 8 presents the same subplots as Figure 3—*All Predictions*, *Best IoU*, and *Best Score*—but includes two additional PSMs: MobileSAM (Tiny) (Zhang & Jiao, 2023) and SAM2.1 (Large) (Ravi et al., 2024). The distributions from these additional models are consistent with the original three, confirming that the findings in Section 3.2 generalize across other PSMs.

Out of the 80 COCO classes, the following 18 are considered *non-uniform*: person, bicycle, car, motorcycle, bus, airplane, train, truck, boat, hot dog, chair, couch, potted plant, bed, dining table, bench, skis, and book.

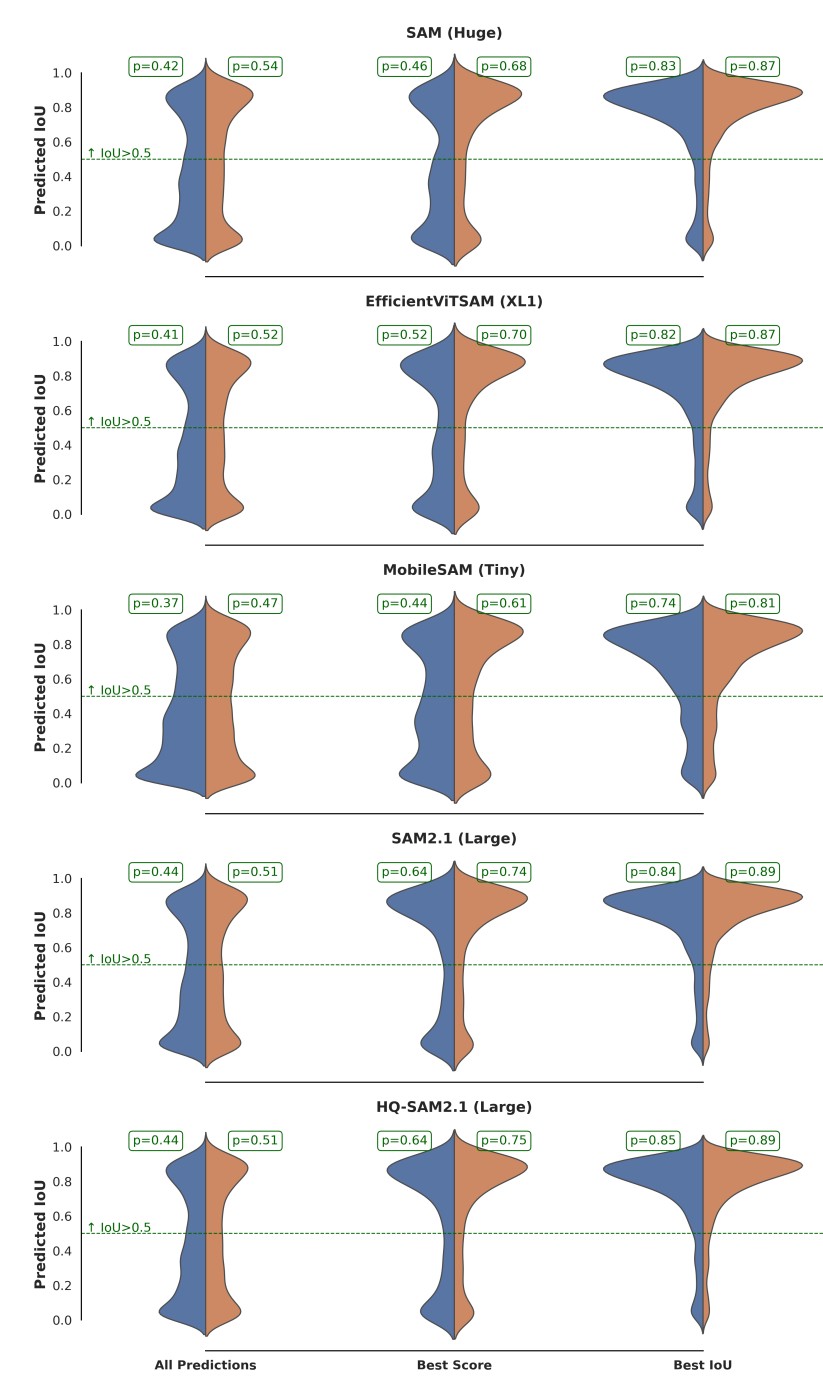

Figure 8: Per-point prediction distributions of different PSMs across IoU values for *uniform* and *non-uniform* objects ($p$ indicates percentage of predictions with IoU > 0.5).

**Effect of Different Scoring Configurations on Per-Instance Distributions.**    To illustrate the effect of each assumption described in Section 4, we presents *Best Score* distributions under different point sampling and scoring configurations using the five PSMs.

1. *Random & Best Raw Score*: $n$ random point prompts within GT masks & original scores.

2. *Random & Best Per-Instance IO Score*: $n$ random point prompts within GT masks & IO scores from instance-level clustering ($n \times m$ embeddings).

3. *TF-OVSS & Best Per-Instance IO Score*: $n$ high-score prompts selected from foreground regions identified with CS (Li et al., 2025) & IO scores from instance-level clustering ($n \times m$ embeddings).

4. *TF-OVSS & Best Per-Image IO Score*: $n$ high-score prompts selected from foreground regions identified with CS (Li et al., 2025) & IO scores from image-level clustering ($n \times m$ embeddings).

Please note that, since the primary focus of this study is to demonstrate the implicit bias of each PSM and the impact of the new scoring method, we consider only a subset of the MS COCO 2017 validation set (Lin et al., 2014), consisting of 500 samples. Among these, we include only the instances for which CS (Li et al., 2025) successfully identified $n = 5$ point prompts. In other words, the PSM tuned for TIDES pipeline may exhibit a slightly different distribution compared to this optimal setting, due to the inclusion of PSM's postprocessing logic under a relaxed configuration (Section A.4).

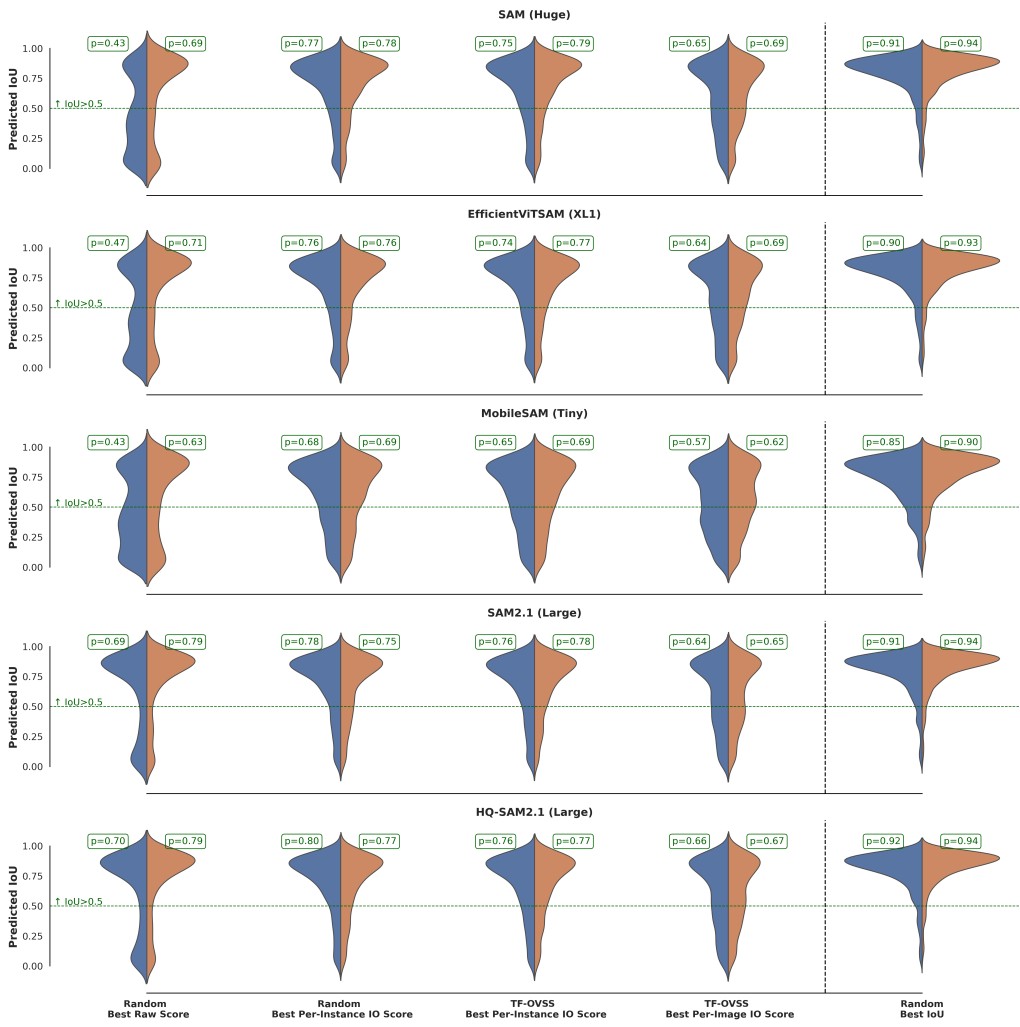

Figure 9: Effect of different point sampling and scoring configurations on the distribution of *Best Score* predictions across PSMs for *uniform* and *non-uniform* objects ($p$ indicates percentage of predictions with IoU > 0.5).

Figure 9 reveals that the changes in distributions resulting from different sampling and scoring configurations exhibit similar patterns across PSMs. Therefore, we describe the identified pattern based on the SAM (Huge) (Kirillov et al., 2023) graph.

From *Random & Best Raw Score* to *Random & Best Per-Instance IO Score*, we observe a 9% increase for *uniform* objects and a substantial 33% increase for *non-uniform* objects, resulting in 78% and 77% of predictions above IoU > 0.5 threshold, respectively. Results from *TF-OVSS & Best Per-Instance IO Score* remain comparable, suggesting that point prompts derived from high patch-text alignment regions do not introduce semantic bias.

When predictions are aggregated across multiple instances (*TF-OVSS & Best Per-Image IO Score*), we observe a 10% decrease for both *uniform* and *non-uniform* objects. Nonetheless, the resulting distributions remain skewed toward higher IoU values (69% for *uniform* and 65% for *non-uniform*), indicating that the misalignment observed in *Random & Best Raw Score* for *non-uniform* objects has been substantially mitigated.

Figure 9 also includes the distributions of *Random & Best IoU* derived from random point prompts with raw scores. The high values of 94% (*uniform*) and 91% (*non-uniform*) represent empirical upper bounds, highlighting the potential for further gains through improved scoring.

## A.4    INFERENCE TIME ANALYSIS

In this section, we analyze TIDES' behavior with respect to PSM postprocessing configurations, aiming to identify the setup that yields the best TIDES performance in practical scenarios.

- *PSM default*: the default postprocessing configuration.
    - *PSM stability score threshold* = 0.95, *PSM box NMS threshold* = 0.70
- *LooseNMS*: only the NMS filtering configuration relaxed.
    - *PSM stability score threshold* = 0.95, *PSM box NMS threshold* = 0.90
- *Balanced*: both the NMS and stability score filtering configurations slightly relaxed.
    - *PSM stability score threshold* = 0.80, *PSM box NMS threshold* = 0.90
- *MinFilter*: both the NMS and stability score filtering configurations relaxed to the extreme.
    - *PSM stability score threshold* = 0.60, *PSM box NMS threshold* = 1.00

Inference latency and corresponding instance segmentation performance are measured by running the pipeline on the same subsampled images used for hyperparameter tuning in Section A.2, but with only the ground-truth classes as the vocabulary. All experiments are performed on a machine equipped with an Intel Core i9-9940X CPU @ 3.30GHz, 125 GB RAM, and a single NVIDIA Quadro RTX 6000 GPU with 24 GB of memory. We report the mean and 95% confidence interval in seconds for the latency, and the AP value for the segmentation performance.

Table 5: Inference latency (in seconds) and segmentation performance (AP) under different PSM configurations. The *Balanced* setting shows a notable performance gain with a moderate increase in latency, and is therefore selected as the default for TIDES.

| PSM | Params. | Inference latency (s) ↓ / segmentation performance (AP) ↑ | | | |
|---|---|---|---|---|---|
| | | *PSM default* | *LooseNMS* | ***Balanced*** | *MinFilter* |
| SAM (Base) | 93.7M | 2.05 ± 0.31 / 21.1 | 2.03 ± 0.33 / 21.4 | 3.51 ± 0.63 / 22.6 | 5.59 ± 0.99 / 24.7 |
| SAM (Huge) | 636.2M | 3.01 ± 0.40 / 26.9 | 3.15 ± 0.42 / 26.8 | 4.48 ± 0.68 / 26.7 | 6.20 ± 1.04 / 27.3 |
| E-SAM (L0) | 34.8M | 1.21 ± 0.06 / 10.5 | 1.22 ± 0.07 / 10.0 | 1.28 ± 0.08 / 13.4 | 1.30 ± 0.08 / 13.5 |
| E-SAM (XL1) | 203.3M | 1.94 ± 0.14 / 26.7 | 2.03 ± 0.15 / 26.6 | 2.72 ± 0.25 / 30.1 | 3.54 ± 0.37 / 29.2 |
| M-SAM (Tiny) | 10.1M | 1.98 ± 0.23 / 21.9 | 2.07 ± 0.26 / 22.1 | 3.54 ± 0.55 / 25.0 | 4.96 ± 0.92 / 24.5 |
| SAM2.1 (Base+) | 80.8M | 1.09 ± 0.05 / 4.8 | 1.10 ± 0.04 / 5.2 | 1.11 ± 0.04 / 7.9 | 1.12 ± 0.04 / 8.8 |
| SAM2.1 (Large) | 224.4M | 1.25 ± 0.05 / 10.6 | 1.23 ± 0.05 / 10.5 | 1.36 ± 0.06 / 20.6 | 1.55 ± 0.08 / 19.7 |
| HQ-SAM2.1 (Large) | 224.7M | 1.74 ± 0.06 / 17.8 | 1.74 ± 0.06 / 17.8 | 1.79 ± 0.07 / 20.4 | 1.97 ± 0.09 / 20.5 |
| Average | - | 1.78 / 17.5 | 1.82 / 17.6 | 2.47 / 20.8 | 3.28 / 21.0 |

As shown in Table 5, TIDES inference time closely correlates with both the size and architectural design of the underlying PSM. Models optimized for speed, such as E-SAM (L0) and M-SAM (Tiny), achieve lower latency with fewer parameters compared to their respective baselines, SAM (Base) and SAM (Huge). Despite having more parameters, SAM2.1 also exhibits improved latency over SAM, owing to its redesigned decoder, optimized prompt encoding, and reduced redundant computation.

As expected, inference latency increases as the filtering conditions are relaxed, due to the higher number of masks processed. Segmentation performance also follows the same pattern, increasing as the filtering conditions are relaxed. However, the performance gain from *Balanced* to *MinFilter* is minimal, showing that a *PSM stability score threshold* of 0.80 and a *PSM box NMS threshold* of 0.90 provide the most return in performance on the investment in inference latency. Therefore, we select this setting as the default for our TIDES pipeline.

## A.5 RESOURCE CONSUMPTION ANALYSIS

To better understand which factors drive resource consumption, we conduct two studies on computational efficiency and memory usage. For all experiments, we evaluate TIDES configured with CS (ViT-B/16) and SAM (Base) on a subset of the LVIS dataset (Gupta et al., 2019). We report per-sample peak GPU memory usage and latency as mean ± standard deviation.

**Sensitivity to the Number of Classes.** Table 6 summarizes how peak GPU memory and latency scale with the number of classes. Both metrics grow steadily as the number of classes increases, showing that TIDES is sensitive to this factor.

Table 6: Sensitivity of TIDES to the number of classes.

| # of Classes | 1 | 25 | 50 | 75 | 100 |
|---|---|---|---|---|---|
| Peak GPU Memory (GB) | 3.3 ± 1.9 | 4.1 ± 0.7 | 5.5 ± 0.9 | 7.5 ± 1.9 | 8.7 ± 2.2 |
| Latency (sec) | 3.0 ± 0.1 | 43.8 ± 2.1 | 85.8 ± 4.1 | 128.7 ± 4.6 | 166.6 ± 12.2 |

**Sensitivity to the Number of Instances.** Table 7 shows how resource usage changes with respect to the average number of instances per image. Here we use a 50-class subset of LVIS. While peak GPU memory grows slightly, latency remains largely unaffected.

Table 7: Sensitivity of TIDES to the number of instances per image.

| Avg. Instances/Image | 2 | 4 | 6 | 8 |
|---|---|---|---|---|
| Peak GPU Memory (GB) | 5.5 ± 0.7 | 5.3 ± 0.5 | 5.9 ± 0.6 | 6.3 ± 0.9 |
| Latency (sec) | 84.5 ± 4.2 | 80.2 ± 4.1 | 84.9 ± 3.5 | 87.3 ± 8.4 |

Improving scalability with respect to both class count and instance density remains an important area to work on for making TIDES more practical and broadly deployable.

## A.6 ADDITIONAL QUANTITATIVE ANALYSIS

To further back up our findings in Section 6, we evaluate the performance of TIDES designed with MobileSAM (Zhang & Jiao, 2023) and SAM2 (Ravi et al., 2024). In addition, we evaluate TIDES linked with smaller version of SAM (Kirillov et al., 2023) (Base) and SAM2.1 (Base+). Our complete evaluation, summarized in Table 8, further confirms that our IO scoring consistently improves performance over raw scores. The evaluation setting is identical to that described in Section 6.

Table 8: Zero-shot TF-OVIS performance of TIDES with different TF-OVSS and PSM combinations on MS COCO 2017. Each includes two variants: Raw and IO scores. "SS perf." denotes the average TF-OVSS performance on standard benchmarks. "SOTA" refers to Zip (Shi & Yang, 2024).

| TF-OVSS (SS Perf.) | PSM | Score | $AR_{10}$ | $AR_{100}$ | $AP_s$ | $AP_m$ | $AP_l$ | $AP_{75}$ | $AP_{50}$ | AP↑ | vs. Naive | vs. SOTA (11.8) |
|---|---|---|---|---|---|---|---|---|---|---|---|---|
| CS (37.2) | SAM (Base) | Raw | 34.9 | 36.1 | 4.6 | 22.4 | 34.4 | 12.3 | 21.7 | 13.0 | | |
| | | IO | 34.1 | 35.0 | 4.6 | 21.5 | 34.1 | 14.2 | 24.8 | 15.2 | +2.2 | +3.4 |
| | SAM (Huge) | Raw | 40.0 | 41.7 | 5.7 | 25.9 | 37.4 | 14.1 | 22.9 | 14.6 | | |
| | | IO | 39.8 | 40.9 | 6.7 | 27.6 | 39.2 | 18.7 | 29.4 | 19.2 | +4.6 | +7.4 |
| | E-SAM (L0) | Raw | 35.6 | 36.3 | 5.3 | 24.1 | 36.5 | 14.6 | 25.2 | 15.6 | | |
| | | IO | 35.5 | 36.1 | 5.4 | 24.4 | 37.5 | 17.5 | 29.3 | 18.5 | +2.9 | +6.7 |
| | E-SAM (XL1) | Raw | 39.4 | 40.3 | 6.4 | 27.9 | 40.3 | 17.8 | 27.3 | 18.2 | | |
| | | IO | 38.5 | 39.3 | 6.9 | 27.6 | 40.8 | 20.5 | 31.2 | 21.0 | +2.8 | +9.2 |
| | M-SAM (Tiny) | Raw | 34.9 | 35.9 | 5.2 | 20.2 | 31.4 | 10.8 | 21.8 | 12.4 | | |
| | | IO | 34.3 | 35.0 | 5.8 | 21.9 | 32.8 | 13.6 | 26.5 | 15.5 | +3.1 | +3.7 |
| | SAM2.1 (Base+) | Raw | 5.8 | 5.8 | 0.0 | 0.6 | 13.3 | 5.5 | 6.5 | 5.4 | | |
| | | IO | 5.8 | 5.8 | 0.0 | 0.6 | 13.2 | 5.4 | 6.5 | 5.4 | 0.0 | -6.4 |
| | SAM2.1 (Large) | Raw | 19.7 | 19.7 | 1.5 | 13.2 | 30.9 | 14.0 | 20.5 | 14.1 | | |
| | | IO | 20.2 | 20.2 | 1.5 | 13.9 | 32.1 | 14.5 | 21.2 | 14.6 | +0.5 | +2.8 |
| | HQ-SAM2.1 (Large) | Raw | 25.6 | 25.7 | 3.4 | 15.4 | 31.8 | 12.3 | 25.4 | 14.6 | | |
| | | IO | 25.9 | 25.9 | 3.5 | 15.8 | 31.8 | 12.5 | 26.3 | 15.0 | +0.4 | +3.2 |
| SCLIP (39.1) | SAM (Base) | Raw | 28.7 | 30.9 | 1.5 | 12.9 | 34.5 | 8.9 | 16.1 | 9.6 | | |
| | | IO | 29.9 | 32.0 | 1.7 | 12.5 | 33.3 | 10.8 | 20.8 | 12.4 | +2.8 | +0.6 |
| | SAM (Huge) | Raw | 36.4 | 39.2 | 2.6 | 20.9 | 40.8 | 13.2 | 20.9 | 13.7 | | |
| | | IO | 35.8 | 37.9 | 3.0 | 21.6 | 41.5 | 17.6 | 26.7 | 18.2 | +4.5 | +6.4 |
| | E-SAM (L0) | Raw | 29.5 | 30.6 | 2.0 | 15.1 | 37.3 | 12.0 | 21.0 | 13.0 | | |
| | | IO | 29.4 | 30.2 | 2.3 | 15.3 | 36.1 | 15.2 | 25.2 | 16.3 | +3.3 | +4.5 |
| | E-SAM (XL1) | Raw | 35.2 | 36.8 | 3.1 | 22.7 | 40.9 | 15.2 | 23.4 | 15.7 | | |
| | | IO | 34.3 | 35.5 | 3.4 | 23.2 | 41.8 | 18.9 | 28.1 | 19.4 | +3.7 | +7.6 |
| | M-SAM (Tiny) | Raw | 31.8 | 33.4 | 2.2 | 15.9 | 33.9 | 9.8 | 19.8 | 11.5 | | |
| | | IO | 31.1 | 32.4 | 2.6 | 16.8 | 34.7 | 12.9 | 24.6 | 14.7 | +3.2 | +2.9 |
| | SAM2.1 (Base+) | Raw | 6.4 | 6.4 | 0.0 | 0.7 | 13.9 | 5.8 | 7.1 | 5.9 | | |
| | | IO | 6.4 | 6.4 | 0.0 | 0.7 | 13.9 | 5.9 | 7.1 | 5.9 | 0.0 | -5.9 |
| | SAM2.1 (Large) | Raw | 19.0 | 19.0 | 0.8 | 10.2 | 32.1 | 13.6 | 20.0 | 13.6 | | |
| | | IO | 18.8 | 18.8 | 0.8 | 10.3 | 32.0 | 13.7 | 20.1 | 13.9 | +0.3 | +2.1 |
| | HQ-SAM2.1 (Large) | Raw | 22.2 | 22.2 | 1.6 | 12.1 | 30.4 | 11.8 | 22.4 | 13.3 | | |
| | | IO | 21.8 | 21.8 | 1.7 | 12.2 | 30.5 | 12.4 | 23.1 | 13.9 | +0.6 | +2.1 |
| SC-CLIP (43.9) | SAM (Base) | Raw | 33.2 | 35.1 | 3.3 | 20.2 | 35.7 | 11.7 | 21.2 | 12.6 | | |
| | | IO | 32.0 | 33.3 | 3.0 | 18.9 | 35.0 | 13.9 | 24.2 | 15.0 | +2.4 | +3.2 |
| | SAM (Huge) | Raw | 40.0 | 42.5 | 4.3 | 25.1 | 40.4 | 14.6 | 23.7 | 15.3 | | |
| | | IO | 38.2 | 40.0 | 4.7 | 25.1 | 39.6 | 18.5 | 28.8 | 19.1 | +3.8 | +7.3 |
| | E-SAM (L0) | Raw | 34.0 | 35.0 | 4.0 | 22.1 | 38.4 | 14.8 | 25.2 | 15.7 | | |
| | | IO | 31.7 | 32.4 | 4.0 | 20.4 | 36.0 | 16.1 | 27.0 | 17.2 | +1.5 | +5.4 |
| | E-SAM (XL1) | Raw | 38.6 | 40.1 | 5.1 | 26.8 | 40.6 | 16.8 | 26.0 | 17.4 | | |
| | | IO | 37.1 | 38.2 | 5.5 | 26.5 | 40.0 | 20.1 | 30.2 | 20.6 | +3.2 | +8.8 |
| | M-SAM (Tiny) | Raw | 34.9 | 36.4 | 4.0 | 19.5 | 33.7 | 11.2 | 22.7 | 13.0 | | |
| | | IO | 33.4 | 34.5 | 4.2 | 19.8 | 33.2 | 13.6 | 26.5 | 15.6 | +2.6 | +3.8 |
| | SAM2.1 (Base+) | Raw | 5.8 | 5.8 | 0.0 | 0.6 | 13.3 | 5.5 | 6.5 | 5.4 | | |
| | | IO | 5.8 | 5.8 | 0.0 | 0.6 | 13.2 | 5.4 | 6.5 | 5.4 | 0.0 | -6.4 |
| | SAM2.1 (Large) | Raw | 19.5 | 19.5 | 1.1 | 12.6 | 31.7 | 14.3 | 20.5 | 14.2 | | |
| | | IO | 19.4 | 19.4 | 1.1 | 12.5 | 32.1 | 14.5 | 20.7 | 14.4 | +0.2 | +2.6 |
| | HQ-SAM2.1 (Large) | Raw | 23.3 | 23.3 | 2.1 | 14.4 | 30.8 | 12.2 | 23.5 | 14.0 | | |
| | | IO | 23.0 | 23.0 | 2.2 | 14.9 | 31.5 | 13.2 | 24.5 | 14.8 | +0.8 | +3.0 |

## A.7 ADDITIAONL QUALITATIVE ANALYSIS

### A.7.1 TIDES WITH OPTIMAL SUBCOMPONENT CONFIGURATIONS

To complement the ablation study in Section 7, we present qualitative results of TIDES (CS (ViT-B/16) & E-SAM (XL1)) configured with the optimal alternative for each subcomponent (Figures 10, 11, and 12). The figures illustrate how improvements to individual subcomponents contribute to overall performance gains, while also providing insights into common failure cases:

1. Instance label incompleteness in ground truth (false negatives):
   Not all instances in the COCO images are labeled, leading to potential underestimation of TIDES' performance in quantitative metrics. TIDES often detects more instances than are annotated, particularly in cluttered scenes (e.g., samples 181666, 238866). While these predictions are semantically valid, they reduce measured precision. For example, a shelf with multiple items labeled as one object may be segmented by TIDES into multiple instances.

2. Non-uniform objects and over-segmentation:
   Objects with distinct subcomponents (e.g., sofas with separate cushions or a person with accessories) may be over-segmented due to their multi-component appearance. For instance, in sample 219578, each sofa cushion is predicted as a separate instance.

3. Class filtering and confusion:
   In the absence of ground-truth semantic supervision, in-context class filtering can produce false positives or misclassifications. For example, in sample 184791, an unlabeled large jug is detected as a bowl.

4. Small object detection:
   While not a primary bottleneck, TIDES shows lower accuracy on small objects, consistent with trends observed in other segmentation models (Table 1).

Throughout our experiments with Zip, we observed that its performance on multi-instance or complex scenes falls significantly below a usable level. High false positives caused by confusion between foreground and background (e.g., samples 238866, 181666) indicate that the main issues stem from an inability to distinguish foreground from background and to correctly identify in-context classes. This highlights the stability and advancements introduced by TIDES.

It is also worth noting that these failures arise from fundamental limitations in Zip's design. Zip focuses on detecting boundary pixels to group non-boundary pixels into instances, which results in aggressive discarding of pixels near instance boundaries. Recovery of these pixels relies on SAM, introducing additional instability in complex scenarios.

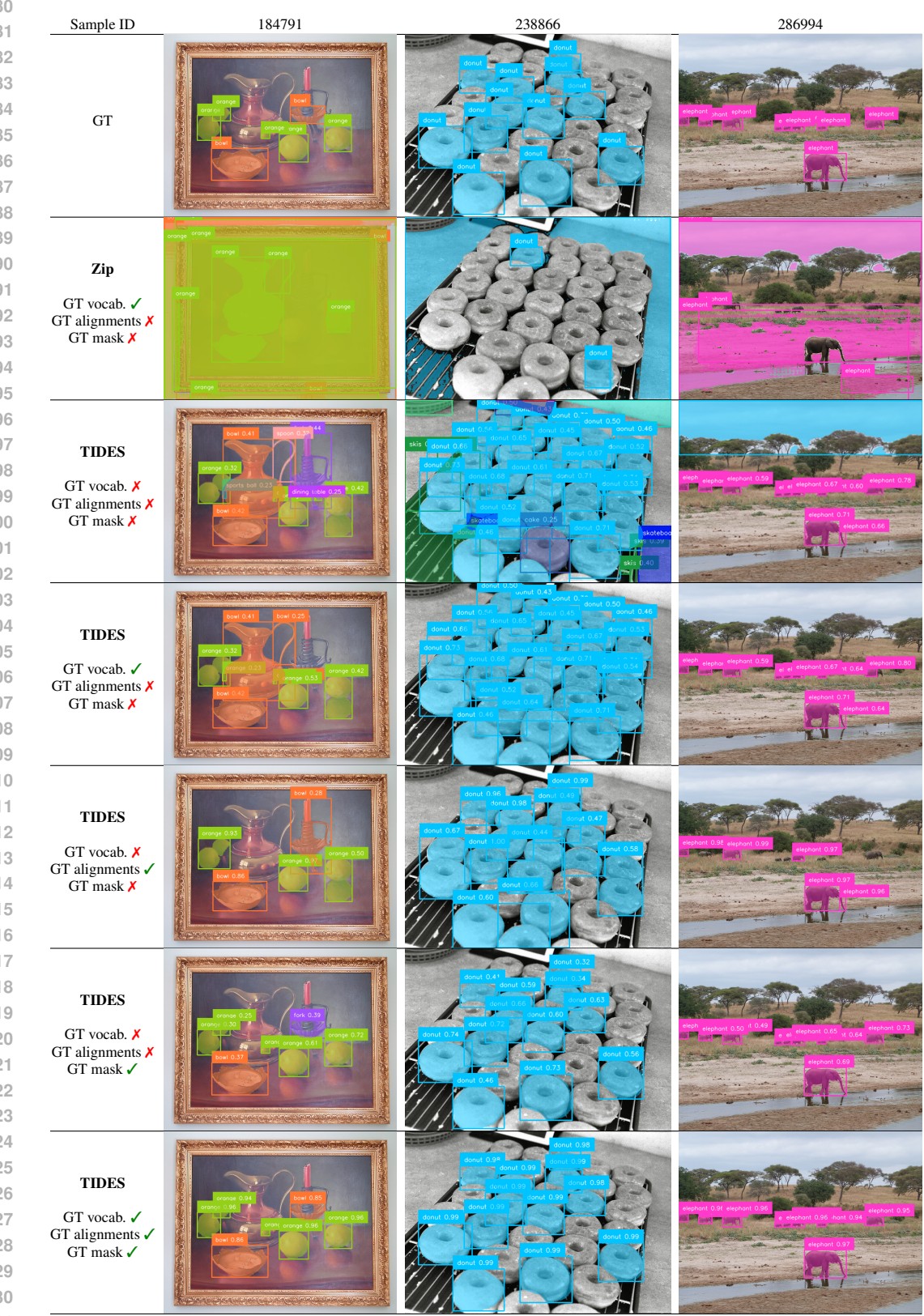

Figure 10: Qualitative results of TIDES, its variants with subcomponents replaced by optimal alternatives, and Zip (Shi & Yang, 2024).

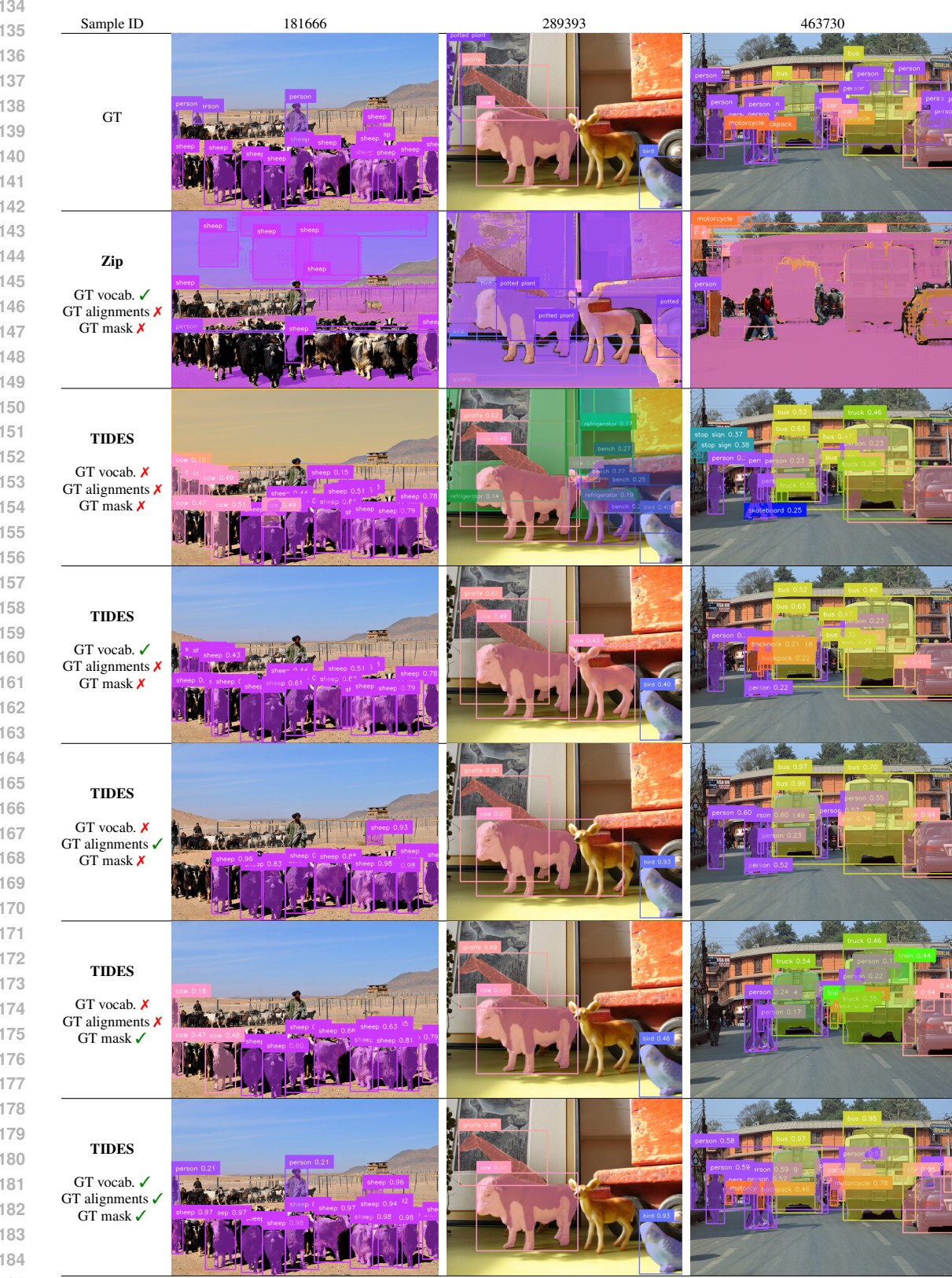

Figure 11: Qualitative results of TIDES, its variants with subcomponents replaced by optimal alternatives, and Zip (Shi & Yang, 2024).

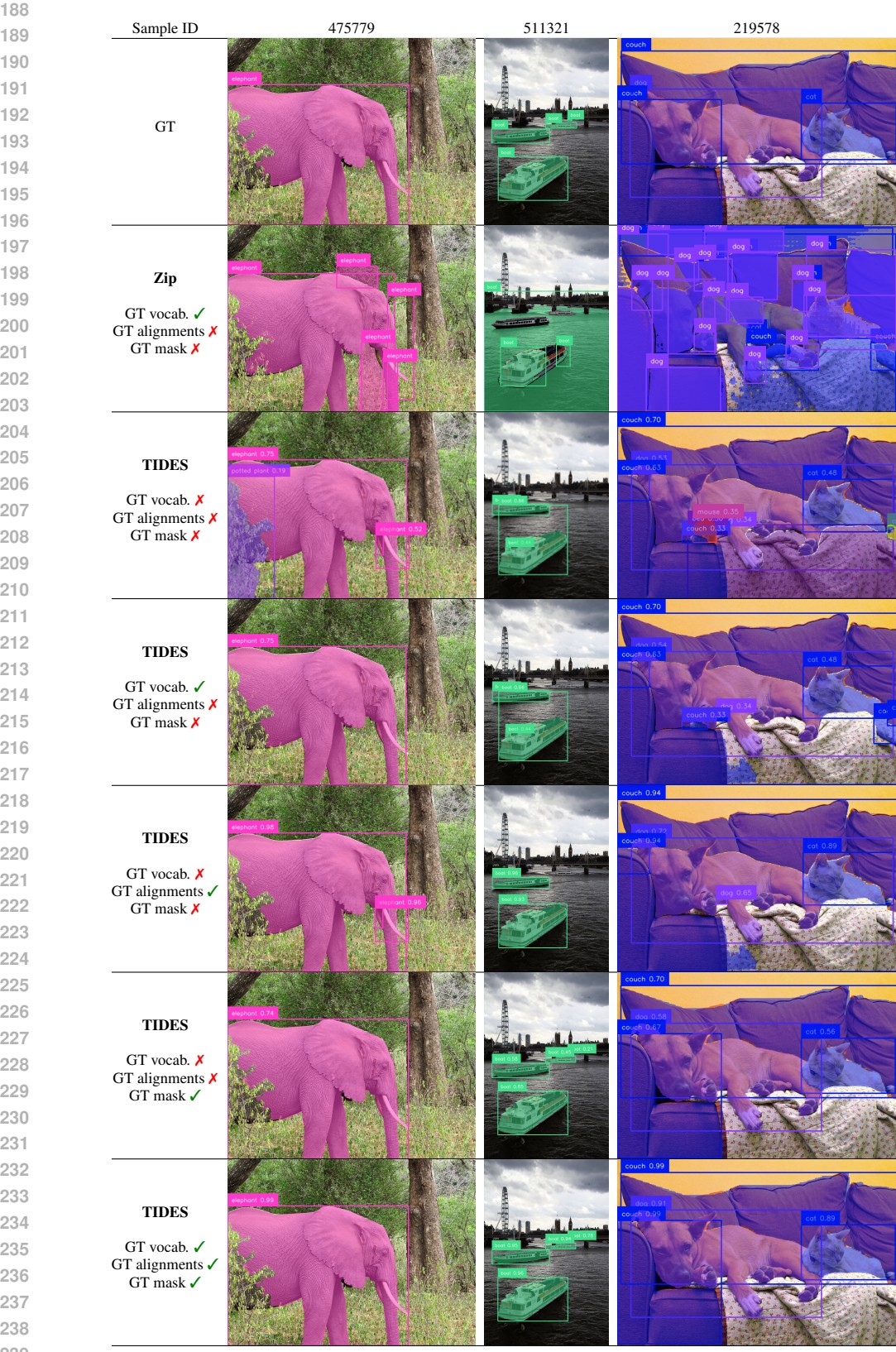

Figure 12: Qualitative results of TIDES, its variants with subcomponents replaced by optimal alternatives, and Zip (Shi & Yang, 2024).

### A.7.2 EFFECT OF PSM PERFORMANCE ON TIDES

The first set of figures (Figure 13, 14, and 15) illustrates how different PSMs affect the final segmentation outputs of TIDES, underscoring the system's reliance on PSM for robust performance. To isolate the impact of the PSM, we use GT class names as prompts and apply the same TF-OVSS method (CS (ViT-B/16)) to derive the patch-text alignment scores.

Based on our comprehensive quantitative and qualitative analysis, EfficientViT-SAM is identified as the most suitable PSM for TIDES. Given HQ-SAM2.1's ability to capture fine-grained details, this difference needs careful examination. Therefore, we summarize the top three masks produced by each PSM in Figure 16, 17, 18, and 19. Our observations reveal that each PSM has distinct characteristics that affect the types of masks retained after its postprocessing logic and TIDES' instance identification. For example, SAM2.1 tends to generate many masks that span multiple objects. This behavior negatively impacts TIDES' clustering process, often causing incorrect masks to be ranked higher than the correct ones. Developing a pipeline that explicitly accounts for the implicit characteristics of each PSM presents a promising direction for future work.

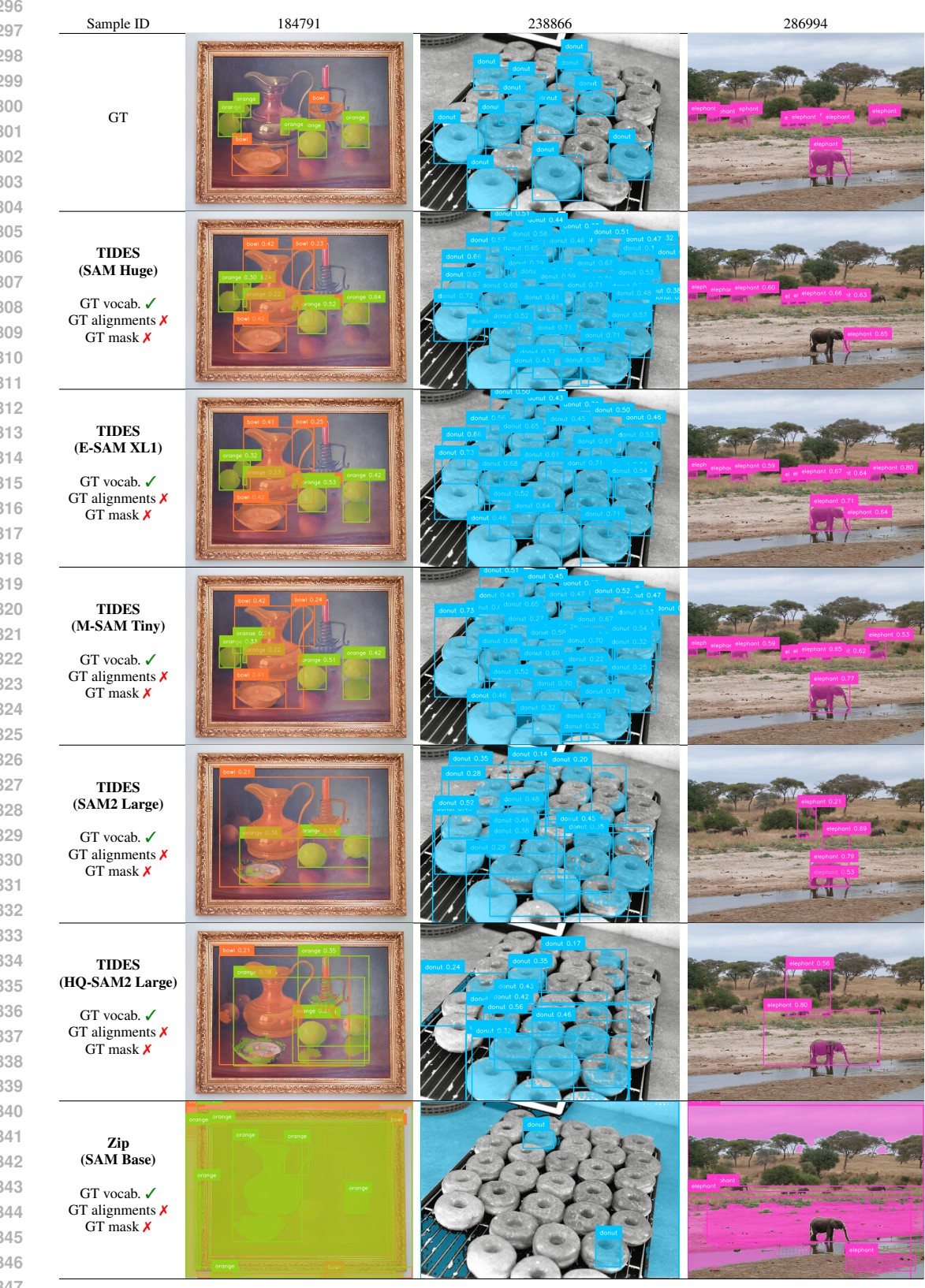

Figure 13: Difference in TIDES performance across underlying PSMs, along with Zip (Shi & Yang, 2024).

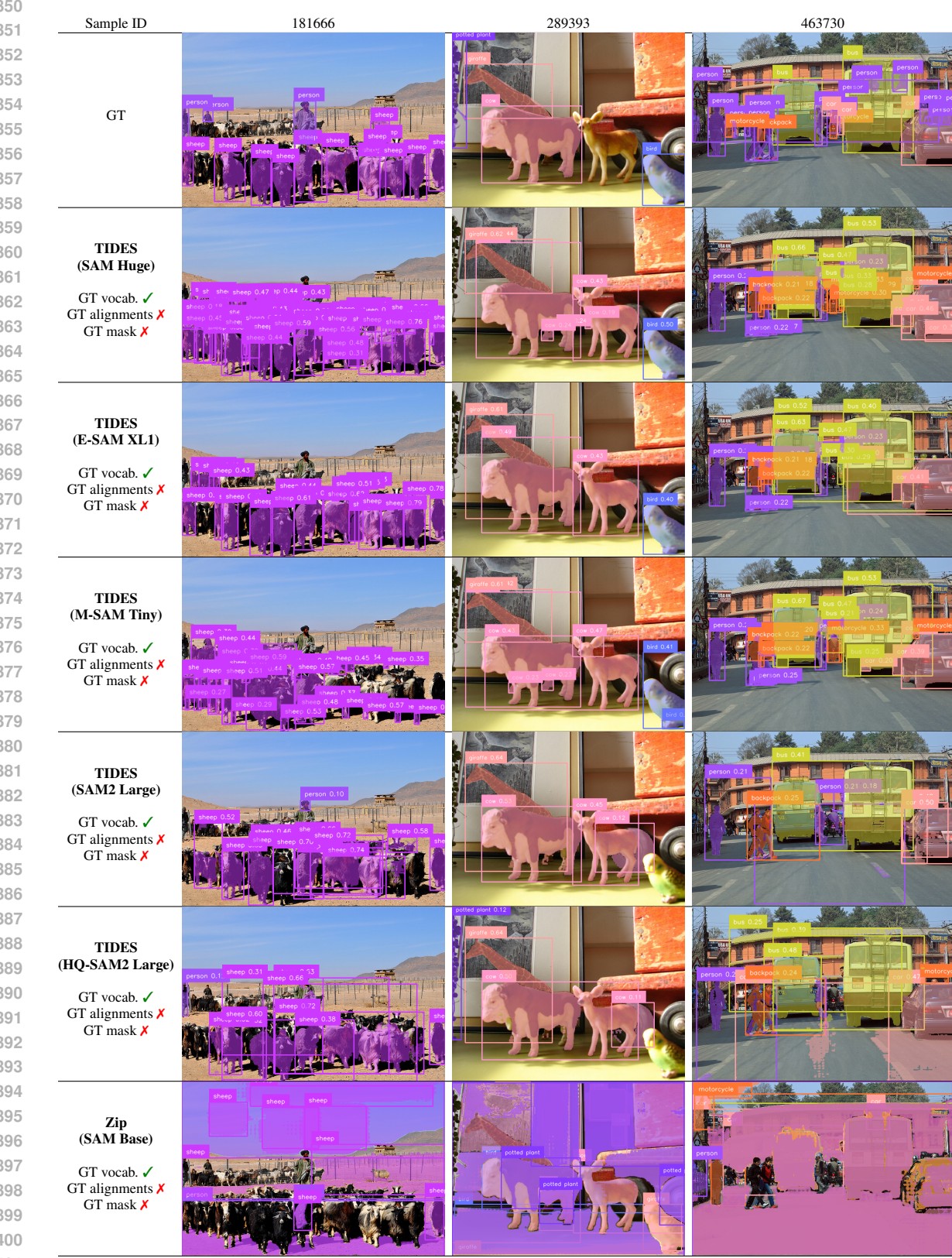

Figure 14: Difference in TIDES performance across underlying PSMs, along with Zip (Shi & Yang, 2024).

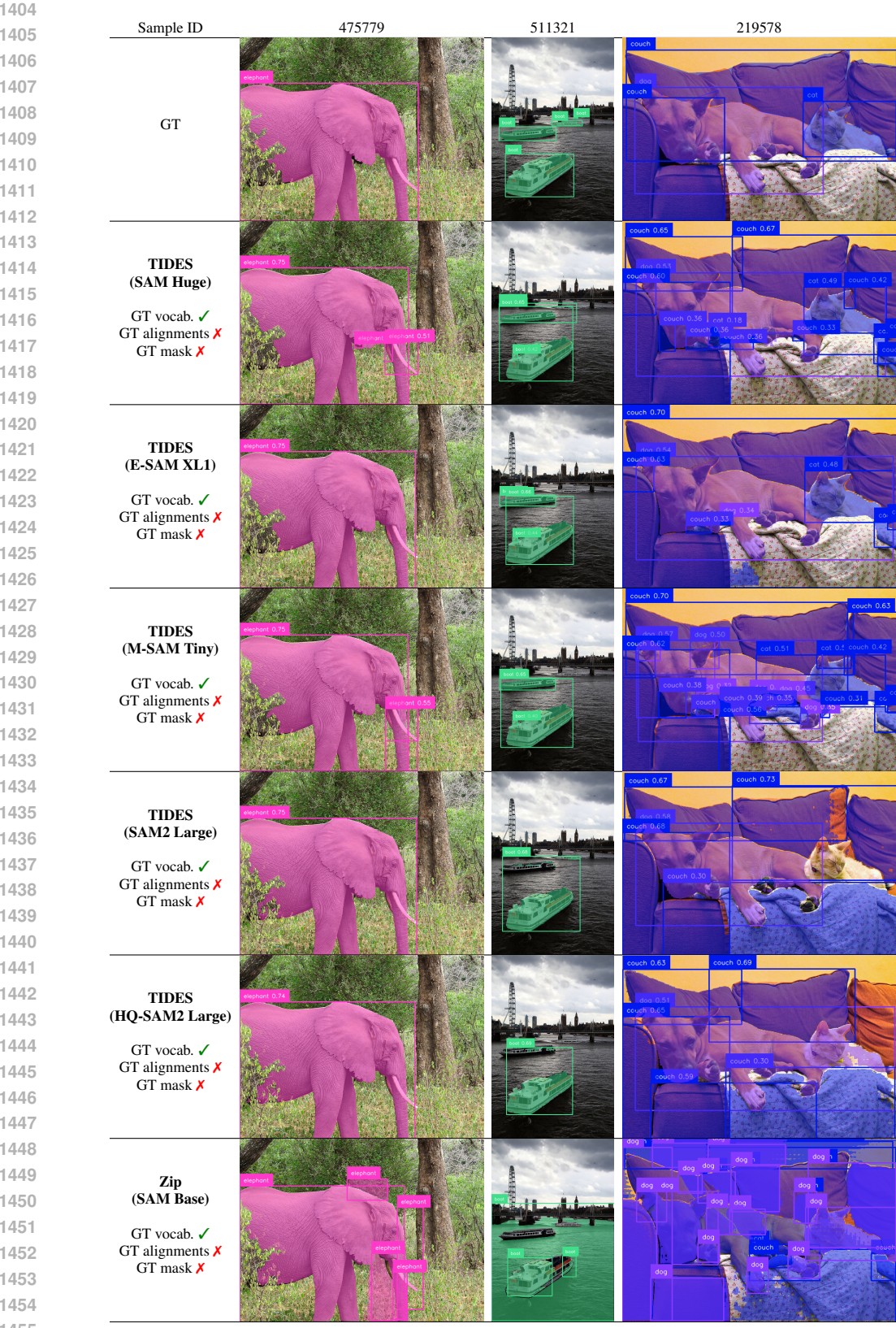

Figure 15: Difference in TIDES performance across underlying PSMs, along with Zip (Shi & Yang, 2024).

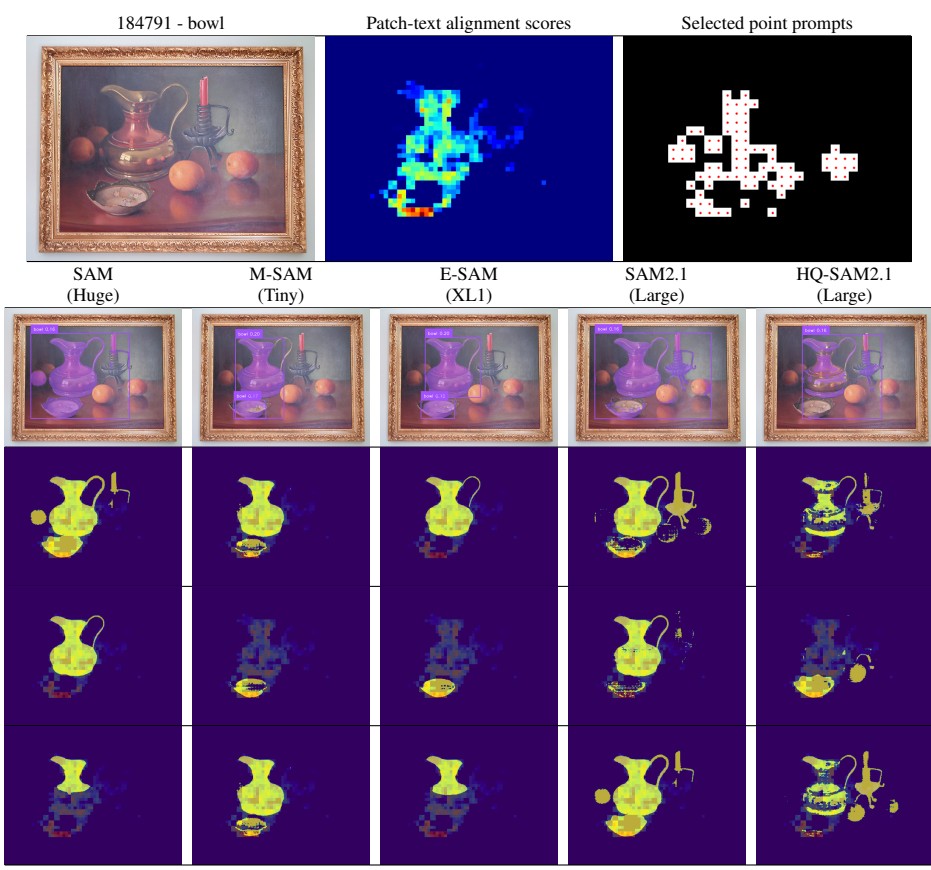

Figure 16: Patch-text alignment scores derived by CS (Li et al., 2025), selected point prompts, and PSM predictions corresponding to the top three TIDES predictions for different PSM.

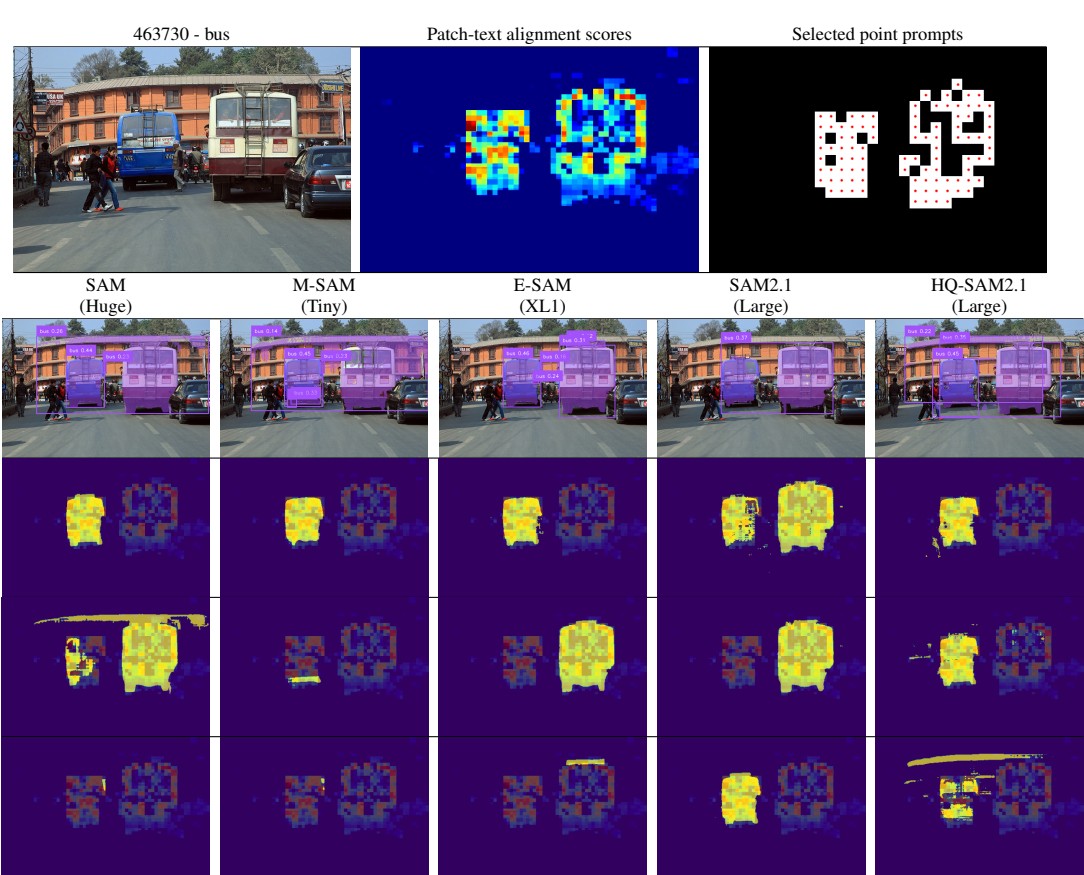

Figure 17: Patch-text alignment scores derived by CS (Li et al., 2025), selected point prompts, and PSM predictions corresponding to the top three TIDES predictions for different PSM.

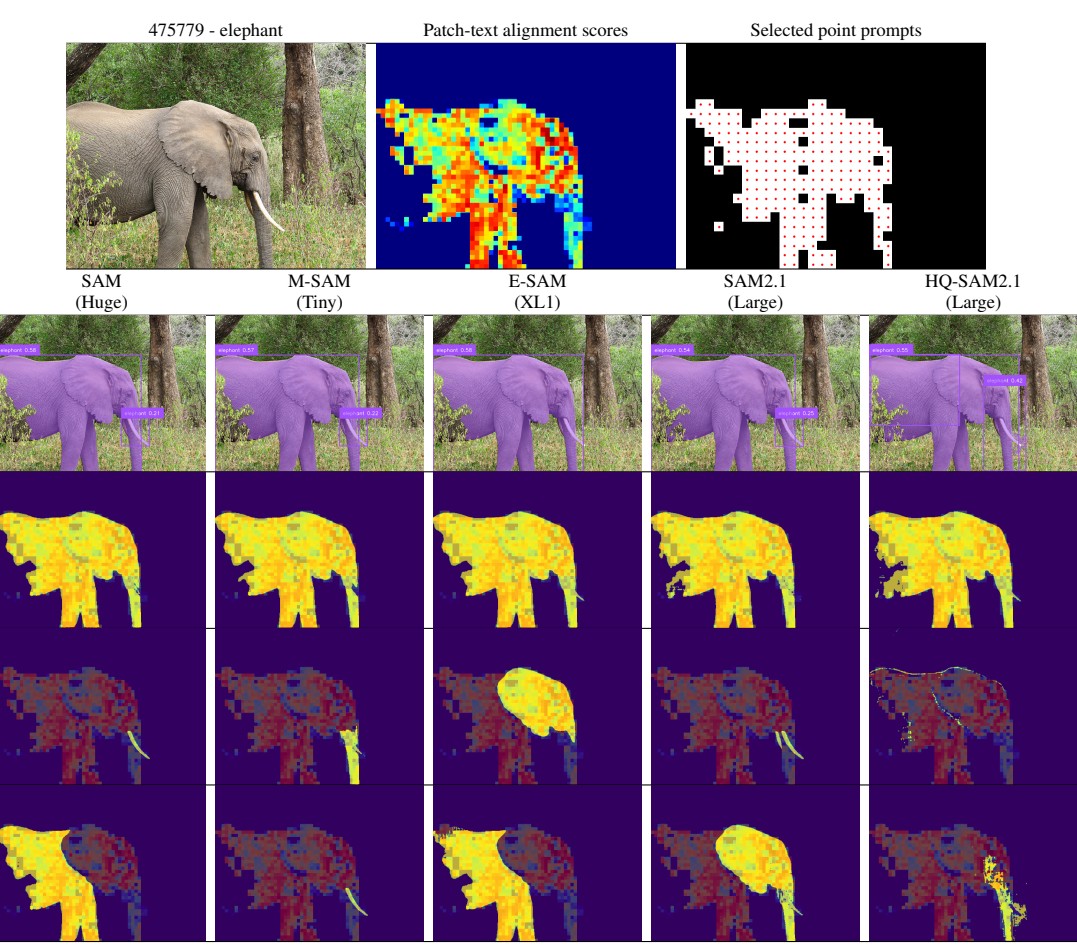

Figure 18: Patch-text alignment scores derived by CS (Li et al., 2025), selected point prompts, and PSM predictions corresponding to the top three TIDES predictions for different PSM.

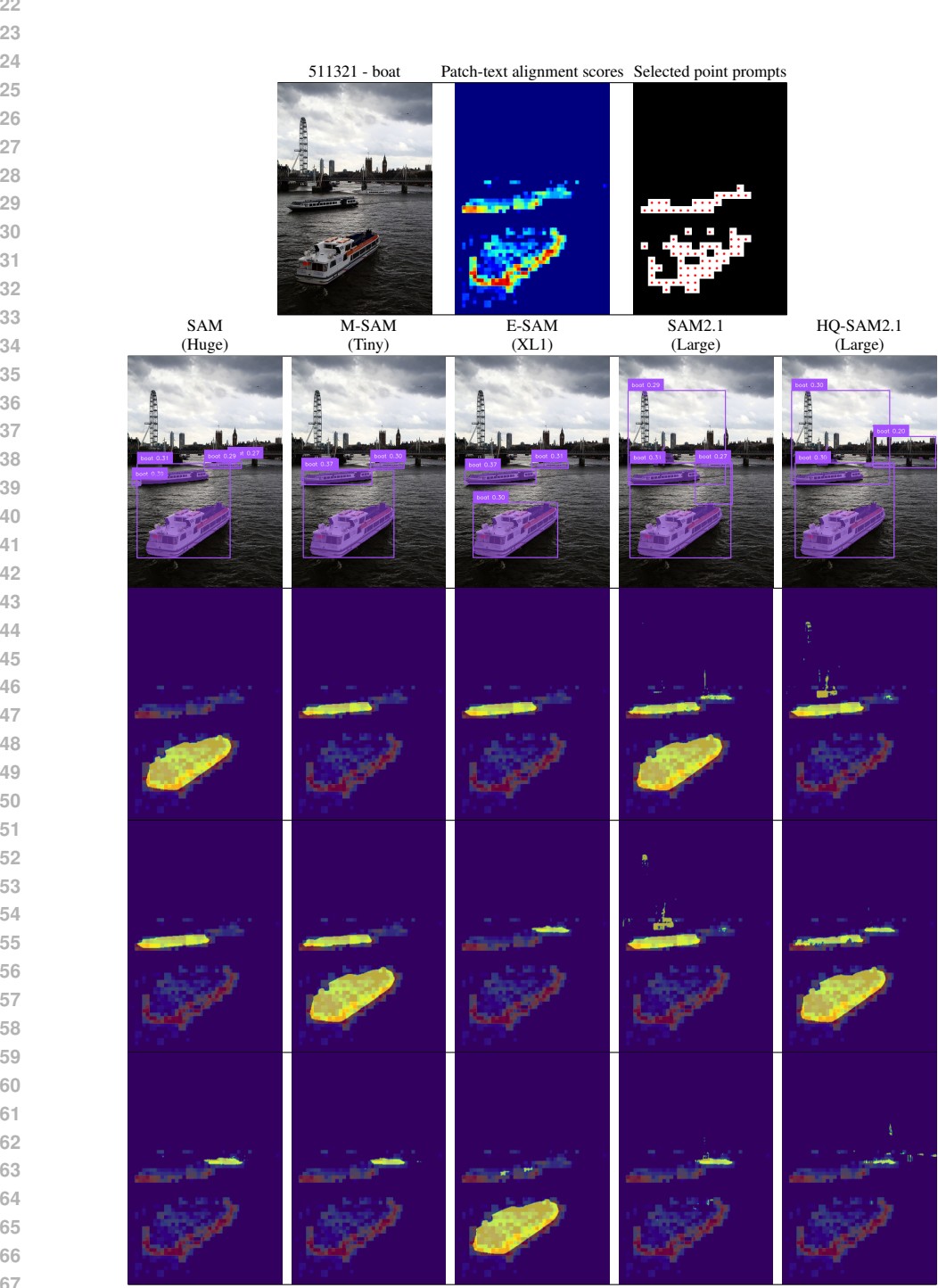

Figure 19: Patch-text alignment scores derived by CS (Li et al., 2025), selected point prompts, and PSM predictions corresponding to the top three TIDES predictions for different PSM.