# OpenReview forum: "TIDES: Training-free Instance Detection from Semantics"
_ICLR.cc/2026/Conference — Submitted to ICLR 2026_

### Official Review · Reviewer_y1ie · 2025-10-28

**Soundness:** 2
**Presentation:** 3
**Contribution:** 3
**Rating:** 6
**Confidence:** 3

**Summary:**

Authors identify a pitfall in current literature: there are lots of work tackling training-free semantic segmentation, but few tackling training-free instance segmentation. The main reason is that backbone models used for TF-OVSS have good understanding of semantics, yet not capable of distinguishing between multiple instances.

**Strengths:**

- Authors correctly identify and tackle a present problem in the computer vision landscape. While most methods aim to solve training-free semantic segmentation, in this work, authors tackle training-free instance segmentation, a much challenging task.

- Authors evaluate their pipeline approach on multiple TF-OVSS and PSM models.

- The method proposed by authors is adaptable to any TF-OVSS model and any PSM model. Furthermore, their performance improves as the future advancements improve the semantics and the instance-level delineation.

- The pipeline solution proposed shows improvements over previous methods, and in general, well-motivated. Overall, I recommend acceptance.

**Weaknesses:**

- Major drawback: There seems to be a lot of engineering hand-crafted tricks and fixed parameters (e.g. 0.2 IoU in line 367, or IoU 0.7 in line 372). There seems to be a lot of tunable hyperparameters which authors show they have a big impact in the downstream AP (Table 4 Appendix). All the ablations and experiments seem to revolve around MSCOCO dataset, possibly overfitting hyperparameter selection to this specific dataset. I believe that building a pipeline, ablating the method, and evaluating the method with a single dataset can be misleading.
- Minor drawback: Pipeline latency grows considerably when the number of classes increases.
- Minor drawback: The method is highly dependent on base vision-language encoder model semantic understanding.

**Questions:**

- Section 4.2. Could authors clarify how are different instances of the same semantic concept differentiated from one another if the histogram distribution is very similar? (e.g. two different apples should yield two very similar histograms/embeddings).

- Isn't randomly selecting 5 points (line 186) a bit unfair? Did authors experiment with sampling points around the center of the instance rather than randomly (which could cause points near the edges of the mask?). Are the same point locations shared among different models (for a fairer comparison)?

- Why is SS Perf value in table 1 much higher than values obtained with Raw and IO?

---

> ### Author Response · Authors · 2025-11-23
>
> Thank you for the thoughtful feedback. Below we address all concerns with clarifications, strengthened explanations, and additional empirical evidence.
>
> ---
>
> ## Weakness 1 - Addressing Evaluation Objectivity Concerns
>
> To address concerns about overfitting to MS COCO, we additionally evaluate TIDES on the "thing-only" subsets of ADE20K and LVIS, using the same evaluation setting as MaskCLIP (ICML 2023).
> We selected MaskCLIP because one of its variants is trained using only mask data without class labels.
>
> Surprisingly, TIDES is found to outperform all mask-trained variants of MaskCLIP on both ADE20K and LVIS.
> ||Training Data||ADE20K|||LVIS||
> |-|-|-|-|-|-|-|-|
> |||AP|AP50|AP75|AP|AP50|AP75|
> |MaskCLIP's CLIP Baseline|COCO Panoptic Masks|3.9|6.0|4.2|4.9|7.2|5.2|
> |MaskCLIP w/o RMA|COCO Panoptic Masks|4.2|6.6|4.4|5.7|8.2|6.1|
> |MaskCLIP|COCO Panoptic Masks & Labels|5.9|9.7|6.2|8.4|12.1|8.8|
> |TIDES (Ours)|N/A|4.4|8.2|4.1|8.8|15.1|8.7|
>
> Furthermore, the table below summarizes TIDES' performance on the Pascal-Part-116 benchmark, which was part of the Open Vocabulary Part Segmentation (OV-PARTS) Challenge (CVPR 2024). It is worth noting that all other available solutions are explicitly trained for part segmentation, whereas TIDES operates entirely without training. Despite this, TIDES achieves performance on unseen objects are comparable to PartCLIPSeg, which remained the state-of-the-art method until very recently.
>
> The default evaluation metric is mIoU, where Oracle-Obj assumes that object masks are available, while Pred-All does not. The harmonic mean IoU (h-IoU) is computed as 2 × (Seen × Unseen) / (Seen + Unseen).
>
> |Method|Venue|Training-free|Pred-All: Seen|Pred-All: Unseen|Pred-All: h-IoU|Oracle-Obj: Seen|Oracle-Obj: Unseen|Oracle-Obj: h-IoU|
> |-|-|-|-|-|-|-|-|-|
> |PartCLIPSeg|NeurIPS'24|No|43.91|23.56|30.67|50.02|31.67|38.79|
> |PartCATSeg (SOTA)|CVPR'25|No|52.62|40.51|45.77|57.49|44.88|50.41|
> |TIDES|N/A|Yes|32.50|25.35|28.49|34.57|26.51|30.01|
>
> ## Weakness 2 & 3 – Analysis of TIDES' Limitations
>
> We are aware of the concerns regarding hyperparameter sensitivity and efficiency. Behaviors such as increased processing time with a larger number of classes or sensitivity to certain hyperparameters are common trade-offs made to improve ease of use. While we believe the ability to operate effectively without training outweighs these costs, we analyze these trade-offs in detail in the Appendix to provide a clearer view of our pipeline.
>
> ## Question 1 - Differentiating Multiple Instances of the Same Semantic Class
>
> Figure 7 in the Appendix visually summarizes TIDES' process for an image containing three elephants. It illustrates how IO scoring and clustering of patch-level semantic alignment score histograms enable stable identification of individual masks in multi-instance scenarios.
>
> The histograms of masks that fully cover each elephant exhibit similar distributions, while masks covering only parts of the elephants show distinct distributions. The distinction between these histograms causes them to be located farther apart in the semantic space, allowing the full-instance masks to form a tight, distinctive cluster. Masks near the cluster center are assigned high scores, and when NMS is applied, overlapping masks for the same instance are removed, leaving only those that correctly cover each full instance.
>
> ## Question 2 - Clarifications on Point Sampling in Section 3
>
> The following summarizes the impact of different sampling methods on the percentage of predictions with IoU > 0.5. We compared random sampling against two other strategies: (1) sampling near the center and (2) sampling the farthest points within the ground-truth masks. The findings are consistent across methods, with Best IoU representing the upper bound and Best IO Score consistently outperforming Best Raw Score.
>
> |Mask Selection Method|Uniform: Random|Uniform: Center|Uniform: FPS|Non-Uniform: Random|Non-Uniform: Center|Oracle-Obj: FPS|
> |-|-|-|-|-|-|-|
> |Best Raw Score|0.47|0.54|0.26|0.36|0.42|0.20|
> |Best IO Score|0.61|0.68|0.33|0.34|0.40|0.19|
> |Best IoU|0.82|0.87|0.50|0.75|0.80|0.45|
>
> The same selected points are used across different mask selection methods to ensure a fair comparison.
>
> ## Question 3 - Clarification on "SS Perf"
>
> "SS Perf." in the table refers to the average TF-OVSS performance on standard semantic segmentation benchmarks, including VOC20, Context59, Cityscapes, and ADE20K. TIDES is evaluated on the COCO instance segmentation dataset, which explains the differences in reported values.

---

### Official Review · Reviewer_xytL · 2025-10-31

**Soundness:** 3
**Presentation:** 3
**Contribution:** 2
**Rating:** 4
**Confidence:** 4

**Summary:**

This paper introduces TIDES, a training-free pipeline for open-vocabulary instance segmentation (OVIS). It tackles the core problem that promptable segmentation models (PSMs) like SAM, while capable of generating accurate instance masks, often assign their highest confidence scores to parts of objects rather than the complete instance. TIDES solves this by combining any semantic segmentation model with any PSM. Its key contribution is the IO scoring method, which re-evaluates all raw masks from the PSM. It converts each mask into a semantic embedding based on a histogram of patch-text alignment scores from the TF-OVSS, clusters these embeddings, and uses the local cluster density as the new IO score. Experiments show that it surpasses previous TF-OVIS SOTA and can adapt many combinations of TF-OVSS models and PSMs.

**Strengths:**

1. The method achieves state-of-the-art results for training-free OVIS, surpassing the previous SOTA by a large margin.
2. The paper provides a clear quantitative analysis of the exact problem with PSMs: their scores are misaligned with human perception of instances, even when they are capable of producing the correct masks. The solution (IO scoring) directly targets this diagnosed weakness. Ablation studies also show consistent improvements with IO scoring.
3. Experiments with multiple semantic models and PSMs suggest the transferability of the proposed approaches.

**Weaknesses:**

1. The pipeline requires a PSM to perform instance segmentation, which requires iterative prompting and merging. This is inefficient and highly dependent on PSM to provide good masks. It cannot improve results when GT masks are given. And it cannot be adapted to class-agnostic entity segmentation models.
2. Though this paper shows good improvements over the previous SOTA, the application scenarios of the proposed method seem restricted. The design of selecting mask candidates makes it infeasible for part segmentation; it is not easy to transfer it for LLM-based reasoning segmentation.

Besides that, this paper has acknowledged or implied the following weaknesses:

3. The pipeline is complex with many hyperparameters and is highly sensitive to hyperparameters (Table 4). Some hyperparameters can cause a significant gap (~20% in AP).
4. The pipeline is inefficient: it takes ~100 seconds for one image when the number of instances and number of classes increase (Table 6 and Table 7).

Overall, I think the approaches come with interesting observations and good motivations. But the problem that this paper tackles is less significant and the setting has limited application scenarios.

**Questions:**

1. It looks like this work is transferrable to open-vocabulary panoptic segmentation. The authors may conduct experiments on such benchmarks and metrics and make some quantitative comparisons to training-based approaches such as MaskCLIP (Ding et al) [1] and ODISE [2]. Moreover, MaskCLIP (Ding et al) [1] already includes a training-free baseline ("w/o RMA" in the paper) that can be compared in a fair setting. This would attract more audience and offer a sense of gaps between training-free and training-based approaches.

[1] Open-Vocabulary Universal Image Segmentation with MaskCLIP, ICML 2023

[2] Open-Vocabulary Panoptic Segmentation With Text-to-Image Diffusion Models, CVPR 2023

---

> ### Author Response · Authors · 2025-11-23
>
> Thank you for the feedback. We address all concerns below.
>
> ---
>
> ## Weakness 1 - Clarifications on PSM Usage
>
> Regarding PSM usage, the input point prompts are batched, and PSM is executed only once per image; no iterative prompting or merging is involved in the process.
>
> The reviewer is correct that TIDES depends on the performance of PSMs. However, it is not limited by their shortcomings. Segmentation failures of PSMs arise from two sources:
>
> 1. Limited mask decoding ability (imprecise boundaries)
> 2. Incorrect semantic understanding (attending to “head” when the prompt is “person”)
>
> As shown in Section 3, PSM proposals often include the correct instance. TIDES primarily addresses the second issue using IO scoring. PSMs implicitly generate multiple proposals and return only the one with the highest internal score. Our IO scoring reorders proposals to match the intended semantic granularity expressed by the input signal. IO scoring is training-free, model-agnostic, and requires no hand-crafted rules, enabling the flexible, plug-and-play design of TIDES that existing methods do not support.
>
> ## Weakness 2 – Practical Advantages of TIDES
>
> Another practical advantage is the ability to (a) reject out-of-context prompts and (b) detect individual instances. Many existing training-free open-vocabulary segmentation models (e.g., OVDiff (ECCV 2024)) assume at least one instance exists in the context and produce a single mask for all instances.
>
> The plug-and-play design enables TIDES to support any semantic granularity, provided appropriate input signals. The table below summarizes TIDES' performance on the Pascal-Part-116 benchmark from the Open Vocabulary Part Segmentation Challenge (CVPR 2024). All other available solutions are training-based, whereas TIDES operates without training. Despite this, TIDES achieves performance comparable to PartCLIPSeg, which remained the SOTA method until very recently.
>
> The default evaluation metric is mIoU, where Oracle-Obj assumes that object masks are available, while Pred-All does not. The harmonic mean IoU (h-IoU) is computed as 2 × (Seen × Unseen) / (Seen + Unseen).
>
> |Method|Venue|Training-free|Pred-All: Seen|Pred-All: Unseen|Pred-All: h-IoU|Oracle-Obj: Seen|Oracle-Obj: Unseen|Oracle-Obj: h-IoU|
> |-|-|-|-|-|-|-|-|-|
> |PartCLIPSeg|NeurIPS'24|No|43.9|23.5|30.6|50.0|31.6|38.7|
> |PartCATSeg (SOTA)|CVPR'25|No|52.6|40.5|45.7|57.4|44.8|50.4|
> |TIDES|N/A|Yes|32.5|25.3|28.4|34.5|26.5|30.0|
>
> ## Weakness 3 & 4 – Analysis of TIDES' Limitations
>
> We are aware of the concerns regarding hyperparameter sensitivity and efficiency. Behaviors such as increased processing time with a larger number of classes or sensitivity to certain hyperparameters are common trade-offs made to improve ease of use. While we believe the ability to operate effectively without training outweighs these costs, we analyze these trade-offs in detail in the Appendix to provide a clearer view of our pipeline.
>
> ## Question 1 – Additional Performance Analysis
>
> ### A. Comparison against MaskCLIP
>
> MaskCLIP's open-vocabulary instance segmentation performance is primarily evaluated on the "thing-only" subsets of ADE20K and LVIS.
> Please note that the two MaskCLIP baselines (CLIP Baseline and MaskCLIP w/o RMA) still require a class-agnostic mask proposal network to be trained with mask data.
>
> Surprisingly, TIDES is found to outperform all mask-trained variants of MaskCLIP on both ADE20K and LVIS.
> ||Training Data||ADE20K|||LVIS||
> |-|-|-|-|-|-|-|-|
> |||AP|AP50|AP75|AP|AP50|AP75|
> |MaskCLIP's CLIP Baseline|COCO Pan. Masks|3.9|6.0|4.2|4.9|7.2|5.2|
> |MaskCLIP w/o RMA|COCO Pan. Masks|4.2|6.6|4.4|5.7|8.2|6.1|
> |MaskCLIP|COCO Pan. Masks & Labels|5.9|9.7|6.2|8.4|12.1|8.8|
> |TIDES|N/A|4.4|8.2|4.1|8.8|15.1|8.7|
>
> MaskCLIP also evaluates their model in a cross-dataset setting (trained on Base evaluate on Target).
> TIDES again outperforms MaskCLIP by 5.9 AP on the unseen classes (generalized setting).
>
> ||Venue|Training-free|Constrained||Generalized|||
> |-|-|-|-|-|-|-|-|
> ||||Base|Target|Base|Target|All|
> |XPM|CVPR'22|No|42.4|24.0|41.5|21.6|36.3|
> |MaskCLIP|ICML'23|No|42.8|23.2|42.6|21.7|37.2|
> |TIDES|N/A|Yes|19.3|30.6|19.2|27.6|21.0|
>
> ### B. TIDES for Panoptic Segmentation
>
> The following table summarizes TIDES' performance on the ADE20K panoptic segmentation benchmark. Although TIDES is training-free, it achieves Panoptic Quality (PQ) comparable to MaskCLIP w/o RMA, which is trained with mask supervision, showing slightly lower Recognition Quality (RQ) but higher Segmentation Quality (SQ). Performance of the two training-based open-vocabulary panoptic segmentation models, ODISE (CVPR2023) and PosSAM (Arxiv 2025) are included for reference.
>
> ||Training Data|PQ|SQ|RQ|
> |-|-|-|-|-|
> |MaskCLIP's CLIP Baseline|COCO Pan. Masks|8.2|53.1|10.5|
> |MaskCLIP w/o RMA|COCO Pan. Masks|9.6|62.5|12.6|
> |MaskCLIP|COCO Pan. Mask & Labels|15.1|70.5|19.2|
> |ODISE|COCO Pan. Mask & Labels|23.4|-|-|
> |PosSAM (SOTA)|COCO Pan. Mask & Labels|29.2|76.3|34.7|
> |TIDES|N/A|9.1|71.5|11.6|

---

### Official Review · Reviewer_VPz9 · 2025-10-31

**Soundness:** 3
**Presentation:** 3
**Contribution:** 2
**Rating:** 4
**Confidence:** 5

**Summary:**

Against the background that vision-language dual encoders (e.g., CLIP) work for training-free open-vocabulary semantic segmentation (TF-OVSS) but fail in training-free open-vocabulary instance segmentation (TF-OVIS) due to poor instance distinction, and promptable segmentation models (PSMs, e.g., SAM) generate instance masks with confidence score bias, the paper addresses the limitations of existing TF-OVIS methods (e.g., Zip’s reliance on specific TF-OVSS, GroundedSAM’s need for instance annotations) by proposing TIDES. TIDES uses an instance-oriented (IO) score leveraging TF-OVSS’s patch-level semantic alignment to re-evaluate PSM masks, enabling flexible combination of any TF-OVSS and PSM without training or instance labels. Experiments on MS COCO zero-shot OVIS show TIDES outperforms naive baselines by 2.7 AP and prior SOTA Zip by 9.2 AP (best 21.0 AP). Its contributions include systematic analysis of PSM’s score bias, the general IO score, flexible TIDES pipeline, and identification of TF-OVIS’s three core challenges.

**Strengths:**

1. The TIDES pipeline is designed with modularity, supporting arbitrary TF-OVSS/PSM combinations. Experiments on MS COCO zero-shot OVIS are comprehensive, with consistent performance gains across 9 model combinations, verifying robustness.
2. The problem definition (TF-OVIS’s challenges) and method logic (IO score’s semantic embedding + clustering) are clearly presented, and Section 7’s component replacement experiments effectively identify TF-OVIS bottlenecks (semantic alignment, class filtering, instance masks).
3. It reveals three core challenges of TF-OVIS (fine-grained semantic alignment, in-context object identification, instance boundary delineation), and provides a new paradigm for fusing pre-trained models for training-free instance segmentation.

**Weaknesses:**

1. Lack of Theoretical Breakthrough: The IO score relies on existing semantic alignment (from TF-OVSS) and clustering (kernel density estimation) techniques—no novel theoretical frameworks or mathematical models are proposed, making the method appear as a “combination of existing tools” rather than an innovative design.
2. Incomplete Experimental Validation: No ablation experiments for TIDES’s core modules (e.g., removing IO score, replacing TF-OVSS with random semantic signals) to verify their necessity; no analysis of performance in special scenarios (small objects, occluded instances, dense scenes) to test robustness.
3. Insufficient Practical Value Assessment: The paper does not compare TIDES with widely used training-free segmentation paradigms (e.g., VLM+SAM) to prove its practical advantage; nor does it analyze efficiency (e.g., inference time, memory cost) to support real-world deployment claims.
3. Weak Visualization: Critical results (e.g., IO score’s improvement on non-uniform objects like cars) lack mask-level comparisons (e.g., PSM’s original mask vs. IO score-selected mask with IoU labels), making it hard to intuitively confirm the method’s effectiveness.

**Questions:**

1.  Since the IO score uses existing semantic alignment and clustering techniques, what unique insights or design principles does it provide beyond “combining TF-OVSS and PSM”? Please explain if there is any theoretical basis for the hyperparameter settings (e.g., \(h=0.5\), density threshold 0.90) rather than empirical selection.
2. Can you supplement ablation experiments to verify the necessity of each TIDES module (e.g., what happens if IO score is replaced with PSM’s original score, or TF-OVSS is removed)? Also, please add performance metrics for small objects (area < 32²) and occluded instances to show robustness.
3. Can you provide mask-level qualitative comparisons (labeled with IoU and confidence scores) between PSM’s original predictions and IO score-optimized results? Also, please visualize semantic embedding clustering (e.g., embedding distribution of different instances) to confirm clustering effectiveness.

---

> ### Author Response · Authors · 2025-11-23
>
> Thank you for the feedback. We address all concerns below.
>
> ## Weakness 1 - Theoretical Foundations of TIDES
>
> The key theoretical insight lies in the formulation of the IO Scoring, which required:
>
> 1. A quantitative analysis of the PSM’s score distribution, and
> 2. A transformation of each mask into a semantically aware, scale-invariant embedding that exposes implicit instance-awareness.
>
> Our method was derived through mathematical analysis (Section 3 & 4) to ensure theoretical soundness. We later observed that KDE naturally achieves the same functional objective, making it a practical and efficient implementation rather than an arbitrary choice.
>
> Being parameter-free and agnostic to both the underlying feature extractor and PSM, TIDES offers a practical advantage as well: plug-and-play functionality. This contrasts with methods that require explicit training for language grounding (e.g., GroundedSAM - relies on GroundingDINO explicitly trained for open-vocabulary detection) or architecture-specific training-free approaches (e.g., ZIP - tied to ResNet-based CLIP).
>
> ## Weakness 2 & Question 2 - Ablation Studies
>
> ### A. Component-Wise Importance Analysis
>
> * The gains from IO scoring are quantified in Table 1. "Raw score" refers to using the original PSM scores instead of IO scoring. The last column ("vs. Raw") yields an average +2.7 AP on average.
> * Ablation study is presented in Section 7, where we replace TIDES subcomponents with their optimal alternatives to assess the importance of each part.
> * The first row of Table 2 shows that directly combining SAM with CLIP results in extremely low performance (2.6 AP).
>
> ### B. Support for Objects of Varying Sizes
>
> Segmentation failures arise from two sources:
>
> 1. Limited mask decoding ability (imprecise boundaries)
> 2. Incorrect semantic understanding (attending to “head” when the prompt is “person”)
>
> As shown in Section 3, PSM proposals often include the correct instance, so TIDES primarily addresses the second issue.
>
> Evaluating TIDES with and without IO scoring (Table 1) shows consistent improvements across all scales (+0.4 AP for small, medium, and large objects on average).
>
> To capture fine-grained objects more effectively, we also process images at three scales using a sliding-window approach (Section 5 & Appendix A.1).
>
> ### C. Understanding TIDES’ Behavior in Varying Detection Scenarios
>
> In Appendix A.7.1, we provide qualitative analyses of TIDES across various scenarios (occluded instances and dense scenes) under each ablation setting discussed in Section 7, along with direct comparisons against the previous SOTA (ZIP). The figures illustrate how improvements to individual subcomponents would contribute to overall performance gains, while also highlighting common failure cases (over-segmentation and imperfect in-context filtering).
>
> ## Weakness 3 - Practical Advantages of TIDES
>
> Table 2 compares TIDES against other training-free instance segmentation methods. A naive combination of SAM and CLIP achieves 2.6 AP, the previous SOTA (ZIP) achieves 11.8 AP, while our method outperforms both with 21.0 AP.
>
> Another practical advantage is the ability to (a) reject out-of-context prompts and (b) detect individual instances. Many existing training-free open-vocabulary segmentation models (e.g., OVDiff (ECCV 2024)) assume at least one instance exists in the context and produce a single mask for all instances.
>
> Inference time analysis is provided in Appendices A.4 and A.5.
>
> ## Weakness 4 - Additional Visualization
>
> Although uploading figures is not supported by OpenReview, the figures in Appendix A.7 should sufficiently address the reviewers’ concerns regarding the visualization.
>
> The Raw vs. IO setting in Experiment 6 quantifies the improvement introduced by IO scoring.
>
> ## Question 1 - Theoretical Foundations of IO Scoring
>
> Regarding the theoretical insight, please refer to Weakness 1.
>
> ### The Impact of Bandwidth Parameters and Density Threshold
>
> Linking distances in deep VL embedding spaces to realistic measures is challenging. However, KDE allows us to interpret the effect of each parameter in the clustering process:
>
> - Bandwidth parameter (h) controls the smoothness of the Gaussian kernel applied to each data point. Larger values produce smoother densities influenced by nearby points, while smaller values focus more on local variations. A small bandwidth parameter helps filter out masks with higher precision.
>
> - Density threshold determines which density peaks are considered distinct clusters. Higher thresholds select only high-confidence peaks, reducing the influence of noise and spurious clusters.
>
> ## Question 3 - Embedding Distribution Visualization
>
> Along with the visualizations discussed in Weakness 4, figure 7 in the Appendix A.1 illustrates TIDES' process for an three elephants image. It shows how IO scoring and clustering of patch-level semantic alignment score histograms enable stable identification of individual masks in multi-instance scenarios.

---

### Official Review · Reviewer_gxEU · 2025-10-31

**Soundness:** 2
**Presentation:** 2
**Contribution:** 2
**Rating:** 4
**Confidence:** 5

**Summary:**

This paper focuses on training-free open-vocabulary instance segmentation (TF-OVIS). It points out that while CLIP-like models have advanced training-free open-vocabulary semantic segmentation (TF-OVSS), they are weak in distinguishing instances. Meanwhile, SAM-like promptable segmentation models (PSMs) can outline instances accurately but often focus on object parts. Additionally, the academic community holds that TF-OVSS cannot be directly applied to instance segmentation. The study proposes an Instance-Oriented (IO) scoring method, which optimizes PSMs' mask selection by combining TF-OVSS semantic scores and increases SAM's correct mask selection probability by 22%. It also develops the TIDES pipeline, which integrates any PSMs and TF-OVSS for training-free instance segmentation.Experiments show that on the MS COCO benchmark, TIDES outperforms baselines by 2.7 AP and surpasses the SOTA method Zip by 9.2 AP. The paper also identifies three key technical challenges of TF-OVIS.

**Strengths:**

1、This paper distinguishes different instances within the same category by introducing a score for evaluating objectness beyond classification scores. Notably, this approach is achieved in a training-free manner, making it more applicable in practical scenarios.

2、The paper leverages the powerful capabilities of current foundation models such as SAM and CLIP to achieve training-free perception. And, the experiments varify the effectiveness of the proposed method.

**Weaknesses:**

1、Proposal masks generated by methods like SAM may represent different granularities. For example, a mask of the "head" is inappropriate when the target is "person", but it constitutes a complete instance for part segmentation (e.g., when the goal is to segment "head"). For training-free segmentation, the model should be capable of hierarchical segmentation, generating different masks based on prompts of varying granularities. This should be validated on datasets such as Pascal Part and ADE20K Part Segmentation.

2、The traditional CLIP-based approach for training-free open-vocabulary semantic segmentation (OVSeg) has a limitation: it requires a predefined list of categories during evaluation, which does not qualify as true open vocabulary. In contrast, generative vision-language model (VLM)-based methods are inherently better suited for achieving open-domain perception. Please discuss the advantages of this approach (referring to the method in question) and VLM-based methods.

3、The paper relies on masks generated by SAM. For hard cases such as small objects—where masks fail to be triggered by prompts—how can this missing issue be addressed?

4、How is the method's inference speed, given that SAM is widely known to generate masks slowly?

**Questions:**

1、Please clarify the advantages of the propsoed framework compared with generative vision-language model (VLM)-based methods, which is more suitable for open-domain.

2、How could the method solves the limitation of mask quality caused by SAM without training or tuning? And how could the method be applied in the segmentation for different granularities.

3、The speed comparison of the propsoed methods and other methods.

---

> ### Author Response · Authors · 2025-11-23
>
> Thank you for the feedback. We address all concerns below.
>
> ---
>
> ## Weakness 1
>
> ### A. Resolving Granularity Issues in PSMs through IO Scoring
>
> The reviewer correctly noted that PSMs alone often produce masks at inconsistent granularities (e.g., “head” vs. “person”). From our comprehensive analysis (Section 3), we find that while PSMs typically generate proposals containing the correct instance, their internal scoring fails to rank them reliably.
>
> This motivates IO scoring, which reorders proposals to match the intended semantic granularity expressed by the input signal (heatmap or segmentation). IO scoring is training-free, model-agnostic, and requires no hand-crafted rules. This enables the flexible, plug-and-play design of TIDES, which existing methods do not support.
>
> ### B. TIDES for Open Vocabulary Part Segmentation (OV-PARTS)
>
> The table below summarizes TIDES' performance on the Pascal-Part-116 benchmark, which was part of the OV-PARTS Challenge hosted at CVPR 2024. It is worth noting that all other available solutions are explicitly trained for part segmentation, whereas TIDES operates entirely without training. Despite this, TIDES achieves performance on unseen objects are comparable to PartCLIPSeg, which remained the state-of-the-art method until very recently.
>
> The default evaluation metric is mIoU, where Oracle-Obj assumes that object masks are available, while Pred-All does not. The harmonic mean IoU (h-IoU) is computed as 2 × (Seen × Unseen) / (Seen + Unseen).
>
> |Method|Venue|Training-free|Pred-All: Seen|Pred-All: Unseen|Pred-All: h-IoU|Oracle-Obj: Seen|Oracle-Obj: Unseen|Oracle-Obj: h-IoU|
> |-|-|-|-|-|-|-|-|-|
> |PartCLIPSeg|NeurIPS'24|No|43.91|23.56|30.67|50.02|31.67|38.79|
> |PartCATSeg (SOTA)|CVPR'25|No|52.62|40.51|45.77|57.49|44.88|50.41|
> |TIDES|N/A|Yes|32.50|25.35|28.49|34.57|26.51|30.01|
>
> ## Weakness 2 - Advantages of TIDES Compared to Existing Alternatives
>
> ### A. Modular Plug-and-Play Design
>
> TIDES' IO scoring depends only on the proposal set and the input signal, so TIDES does not rely on a specific feature extractor or PSM.
> This contrasts with methods that require explicit training for language grounding (e.g., GroundedSAM, which relies on GroundingDINO explicitly trained for open-vocabulary detection) or architecture-specific training-free approaches (e.g., ZIP, which is tied to ResNet-based CLIP).
>
> ### B. True Open-Vocabulary Capability via In-Context Class Filtering
>
> We agree with the reviewer that prior TF-OVSS/OVSeg pipelines are not truly open-vocabulary, as they rely on predefined label lists.
> To address this, TIDES evaluates the semantic alignment of each class independently against a background class, without depending on other classes’ signals.
> This in-context class filtering (Section 5) enables true open-vocabulary functionality without predefined lists.
>
> ### C. Advantage of TIDES over Generative VLM-Based Segmentation Methods
>
> Among generative VLM-based segmentation models, only a few operate in a fully training-free manner, such as OVDiff (ECCV 2024) and Segment Anyword (PMLR 2025). While they support open-vocabulary segmentation, they (1) assume at least one instance exists in the context and/or (2) provide a single mask for all instances, i.e., the semantic segmentation setting. Furthermore, these methods incur substantial inference costs:
>
> - OVDiff: ~110 s per class per image
> - Segment Anyword: ~470 s per image (1100 steps) or ~28 s with 50 steps
>
> In contrast, TIDES, as a non-generative method, runs in approximately 3 s per image while providing distinct masks for each target instance, all without any training (Appendix A.4).
>
> ## Weakness 3 - Support for Objects of Varying Sizes
>
> Segmentation failures in PSM-only setting arise from two sources:
>
> 1. Limited mask decoding ability (imprecise boundaries)
> 2. Incorrect semantic understanding (e.g., attending to “head” when the prompt refers to “person”)
>
> As shown in Section 3, PSM proposals often include the correct instance, so TIDES primarily addresses the second issue. TIDES primarily addresses the second issue using IO scoring.
>
> Evaluating TIDES with and without IO scoring (Table 1) shows consistent improvements across all scales (+0.4 AP for small, medium, and large objects on average).
>
> To capture fine-grained objects more effectively, we also process images at three scales using a sliding-window approach (Section 5 & Appendix A.1).
>
> ## Weakness 4 - Inference Time Analysis
>
> Inference time analysis is provided in Appendices A.4 and A.5.
>
> ---
>
> ## Question 1 - Advantage of TIDES over Generative VLM-Based Segmentation Methods
>
> Please refer to our response provided for Weakness 2.C.
>
> ## Question 2 - Support for Objects of Varying Sizes & Semantic Granularities
>
> Please refer to our response to Weaknesses 1.B and 3 regarding TIDES’ advantages in detecting finer semantic granularities and small objects.
>
> ## Question 3 - Inference Time Analysis
>
> Inference time analysis is provided in Appendices A.4 and A.5.

---

### Meta-Review · Area_Chair_ScRk · 2026-01-06

**Summary:**

Authors identify and tackle an interesting and useful problem of training-free instance segmentation, while most methods only aim to solve training-free semantic segmentation. However, the solution is viewed as highly engineered with limited novelty. The consensus supports rejection due to the pipeline being "highly engineered" with excessive complexity and reliance on numerous hyperparameters, which undermines its practical utility.

**Reviewer Concerns:**

Although overfitting concerns were alleviated by new ADE20K/LVIS results, the primary critique regarding the method's reliance on "hand-crafted tricks" persists. Reviewers xytL and y1ie note that high sensitivity to hyperparameters (Table 4) creates significant performance gaps, limiting practical robustness.

**Reviewer Scores:**

Reviewers gxEU, VPz9, and xytL would likely retain their scores (4), as the theoretical weakness and complexity were not fundamentally fixed by additional experiments. Reviewer y1ie would likely lower their score (6 $\to$ 5), as the drawbacks regarding engineering heuristics and hyperparameter sensitivity outweigh the motivation of solving an interesting and hard problem.

---

### Decision · Program_Chairs · 2026-01-26

Reject